# Oncogene expression from extrachromosomal DNA is driven by copy number amplification and does not require spatial clustering in glioblastoma stem cells

Karin Purshouse[1,2], Elias T Friman[1], Shelagh Boyle[1], Pooran Singh Dewari[2], Vivien Grant[2], Alhafidz Hamdan[2], Gillian M Morrison[2], Paul M Brennan[2,3], Sjoerd V Beentjes[1,4], Steven M Pollard[2]*, Wendy A Bickmore[1]*†

[1]MRC Human Genetics Unit, Institute of Genetics and Cancer, The University of Edinburgh, Edinburgh, United Kingdom; [2]Centre for Regenerative Medicine and Cancer Research UK Edinburgh Centre, Institute for Regeneration and Repair, The University of Edinburgh, Edinburgh, United Kingdom; [3]Centre for Clinical Brain Sciences, University of Edinburgh, Edinburgh, United Kingdom; [4]School of Mathematics, University of Edinburgh, Edinburgh, United Kingdom

**\*For correspondence:**
steven.pollard@ed.ac.uk (SMP);
wendy.bickmore@ed.ac.uk
(WAB)

†Lead contact

**Competing interest:** The authors declare that no competing interests exist.

**Abstract** Extrachromosomal DNA (ecDNA) are frequently observed in human cancers and are responsible for high levels of oncogene expression. In glioblastoma (GBM), ecDNA copy number correlates with poor prognosis. It is hypothesized that their copy number, size, and chromatin accessibility facilitate clustering of ecDNA and colocalization with transcriptional hubs, and that this underpins their elevated transcriptional activity. Here, we use super-resolution imaging and quantitative image analysis to evaluate GBM stem cells harbouring distinct ecDNA species (*EGFR, CDK4, PDGFRA*). We find no evidence that ecDNA routinely cluster with one another or closely interact with transcriptional hubs. Cells with *EGFR*-containing ecDNA have increased *EGFR* transcriptional output, but transcription per gene copy is similar in ecDNA compared to the endogenous chromosomal locus. These data suggest that it is the increased copy number of oncogene-harbouring ecDNA that primarily drives high levels of oncogene transcription, rather than specific interactions of ecDNA with each other or with high concentrations of the transcriptional machinery.

## Editor's evaluation

This study convincingly shows that, in contrast to recent reports, the transcriptional output of oncogenes carried on extrachromosomal DNA (ecDNA) in glioblastoma cell lines is driven by the copy number of the ecDNA, rather than their spatial localization into transcriptional hubs. This study is relevant to researchers interested in nuclear function, particularly transcriptional organization within malignant cells.

## Introduction

Glioblastoma (GBM) is characterized by intra-tumoural heterogeneity and stem cell-like properties that underpin treatment resistance and poor prognosis (*Bulstrode et al., 2017*; *Suvà et al., 2014*). GBM is divided into distinct transcriptional subtypes that span a continuum of stem cell/developmental

and injury response/immune evasion cell states (*Richards et al., 2021*; *Verhaak et al., 2010*; *Wang et al., 2021*). Genetically, activation or amplification of *EGFR* (chr7) is altered in almost two-thirds of GBM (*Brennan et al., 2013*). Other commonly amplified genes include *PDGFRA* (chr4), *CDK4*, *MDM2* (chr12), *MET*, and *CDK6* (chr7) with multicopy extrachromosomal DNA (ecDNA) considered a major mechanism for oncogene amplification (*Brennan et al., 2013*; *Kim et al., 2020*; *Snuderl et al., 2011*; *Szerlip et al., 2012*).

Although a long-recognized feature of cancer (*Cox et al., 1965*), ecDNA are particularly common in GBM, with 90% of patient-derived GBM tumour models harbouring ecDNA (*Turner et al., 2017*). However, there is much broader interest in mechanisms of ecDNA function across many solid tumours, as ecDNA enable rapid oncogene amplification in response to selective pressures, and have been shown to correlate with poor prognosis and treatment resistance (*Kim et al., 2020*; *Nathanson et al., 2014*; *Vicario et al., 2015*). EcDNA are centromere-free DNA circles of around 1–3 Mb in size that frequently exist as doublets (double minutes), but also as single elements (*Hamkalo et al., 1985*; *Verhaak et al., 2019*; *Vogt et al., 2004*). EcDNA can be composed of multiple genetic fragments generated as a result of chromothripsis (*Gibaud et al., 2010*; *Shoshani et al., 2021*; *Rosswog et al., 2021*). Although ecDNA were previously identified in 1.4% of cancers, more recent studies have shown their prevalence to be significantly higher (*Fan et al., 2011*; *Kim et al., 2020*; *Turner et al., 2017*). EcDNA can lead to oncogene copy number being amplified to >100 in any given cell, with significant copy number heterogeneity between cells (*Lange et al., 2022*; *Turner et al., 2017*). Freed from the constraints imposed by being embedded within a chromosome, ecDNA have spatial freedom and can adapt to targeted therapeutics (*Lange et al., 2022*; *Nathanson et al., 2014*). For example, the *EGFR* variant *EGFRvIII* (exon 2–7 deletion) is found on ecDNA, and is associated with an aggressive disease course and resistance mechanisms against EGFR inhibitors (*Brennan et al., 2013*; *Inda et al., 2010*; *Nathanson et al., 2014*; *Turner et al., 2017*).

As well as their resident oncogenes, ecDNA also harbour regulatory elements (enhancers) required to drive oncogene expression (*Morton et al., 2019*; *Zhu et al., 2021*). Consistent with this, ecDNA have been found to have regions of largely accessible chromatin (assayed by ATAC-seq), indicative of nucleosome displacement by bound transcription factors, and to be decorated with histone modifications associated with active chromatin (*Wu et al., 2019*). Transcription factors densely co-bound at enhancers have been suggested to nucleate condensates or 'hubs' (*Cho et al., 2018*; *Rai et al., 2018*; *Strom and Brangwynne, 2019*), enriched with key transcriptional components such as mediator and RNA polymerase II (PolII) to drive high levels of gene expression (*Cho et al., 2018*; *Chong et al., 2018*; *Sabari et al., 2018*). Given the colocation of enhancers and driver oncogenes on ecDNA, it has therefore been suggested that ecDNA cluster together in the nucleus, driving the recruitment of a high concentration of RNA PolII and creating ecDNA-driven nuclear hubs that in turn enhance the transcriptional output from ecDNA (*Adelman and Martin, 2021*; *Hung et al., 2021*; *Yi et al., 2021*; *Zhu et al., 2021*).

Here, using super-resolution imaging of primary GBM cell lines, we find that ecDNA are widely dispersed throughout the nucleus and we find neither evidence of ecDNA clustering together nor any significant spatial overlap between ecDNA and large PolII hubs. As expected, we show that expression from genes on ecDNA, both at mRNA and protein level, correlates with ecDNA copy number in the tumour cell lines. However, transcription of genes present on each individual ecDNA molecule appears to occur at a similar efficiency (transcripts per copy number) to that of the equivalent endogenous chromosomally located gene. These data suggest that it is primarily the increased copy number of ecDNA in GBM stem cells, and not a specific property of nuclear colocalization, that drives the increased transcriptional capacity of their resident oncogenes.

## Results
### EcDNA are more frequently located centrally in the nucleus in GBM stem cells

We characterized two GBM-derived glioma stem cell (GSC) primary cell lines containing multiple *EGFR*-harbouring ecDNA (ecEGFR) populations (GCGR-E26 and GCGR-E28, referred to here as E26 and E28). Whole genome sequencing (WGS) analysis using Amplicon Architect (*Deshpande et al., 2019*) indicated that E26 ecDNA harbour an *EGFRvIII* (exon 2–7 deletion), and E28 have a

subpopulation of ecDNA with *EGFR* exon 7–14 deleted (*Figure 1A*). The presence of *EGFR* on ecDNA was confirmed by DNA FISH on metaphase spreads (*Figure 1B and C*). E26 harboured more ecDNA per cell than E28 (*Figure 1D*), with approximately 10% of metaphases also indicating the presence of a chromosomal homogeneously staining region (HSR) (*Figure 1B*; arrow). Endogenous *EGFR* is located on human chromosome 7, and metaphase spreads of the two tumour lines showed 3–6 copies of chromosome 7 in E26 and frequently 3 copies in E28 (*Figure 1E*).

Human chromosomes have non-random nuclear organization, with active regions preferentially located towards the central regions of the nucleus (*Boyle et al., 2001*; *Croft et al., 1999*). We sought to determine the nuclear localization of ecDNA in GBM cell lines as compared with the endogenous chromosomal *EGFR*. DNA FISH for chromosome 7 and *EGFR* in nuclei from human fetal neural stem cells (NSCs) confirmed the trend for human chromosome 7 to be generally found towards the periphery of the nucleus (*Boyle et al., 2001 Figure 1F and G*, *Figure 1—figure supplement 1*, *Figure 1—source data 1*). Signal intensity analysis for equally sized bins eroded from the edge to the centre of each nucleus indicated that chromosome 7 and *EGFR* signal intensity were preferentially located towards the nuclear periphery in each cell line (*Figure 1—figure supplement 1*, *Figure 1—source data 1*). Even once chromosome 7 signal was accounted for, *EGFR* DNA FISH signal was still highest at the periphery of NSC nuclei and lowest in the central regions (p<0.0001) (*Figure 1G*), likely reflecting the centromere proximal localization of endogenous *EGFR* on chromosome 7 (*Carvalho et al., 2001*). This radial organization was still significant (p=0.012), but much less marked, in E28 cells which have on average a modest number of EGFR ecDNA compared to endogenous copies (*Figure 1D*). In E26 cells, which have a very high copy number of ecDNA, this preference for a more peripheral localization is lost (p=0.06). These data suggest that, freed of the constraints on nuclear localization imposed by human chromosome 7, *EGFR* genes located on ecDNA can access more central regions of the nucleus.

## *EGFR*-containing ecDNA in GBM stem cells do not cluster in the nucleus

It has been suggested that ecDNA cluster into 'ecDNA hubs' within nuclei of cancer cells, including for *EGFRvIII*-containing ecDNA in other GBM cell lines (HK359 and GBM39) (*Hung et al., 2021*; *Yi et al., 2021*). We sought to quantify this using our E26 and E28 GBM cells with a single oncogene-harbouring ecDNA population (*EGFR* variant amplicons). Previous studies exploring genomic loci proximity and contact domains (*Williamson et al., 2016*; *Williamson et al., 2019*; *Hansen et al., 2021*), and the proximity of super-enhancers to BRD4/MED1 puncta (*Sabari et al., 2018*), would suggest that ecDNAs clustering together at a transcriptional hub should be located within ~200 nm or less of each other. We used 3D image-based analysis of the *EGFR* DNA FISH signals (*Figure 2A*) to determine if there is clustering of ecDNA. The relative frequency of all shortest *EGFR-EGFR* distances per nucleus did not suggest frequent ecDNA-ecDNA interactions at ≤200 nm in either cell line (*Figure 2B*, *Figure 2—figure supplement 1A*). The mean shortest interprobe distances per nucleus were also not suggestive of close interactions, with no values <500 nm (*Figure 2—figure supplement 1B, C*; *Figure 2—source data 1*). The single shortest interprobe distance per nucleus was also larger (0.24 µm, E26; 0.25 µm, E28) than would be expected if there were clustering of ecDNA in the close proximity required for coordinated transcription in hubs (*Figure 2—figure supplement 1D, E*; *Figure 2—source data 1*).

The analysis above quantified distances between FISH hybridization signals but does not determine whether there is a non-random distribution of foci in the nuclei at distances in keeping with transcription hubs. We therefore used 3D Ripley's K function to determine the observed spatial pattern of the foci in each nucleus and compared this with a random null distribution of 10,000 simulations of the same number of foci in the same volume. We powered this to identify any significant clustering at each radius in 0.1 µm increments between 0.1 and 1 µm (examples of E26 and E28 nuclei and their corresponding Ripley's K function in *Figure 2C*). The E26 cell line had some nuclei with significant non-random distribution of ecDNA, but only at ≥400 nm radial distances, and E28 only had occasional nuclei with significant non-random distribution of ecDNA at ≥700 nm (*Figure 2D*). We repeated this analysis, reducing the focus spot size from 300 to 150 nm diameter to ensure no small FISH foci were omitted that might skew our analysis. No significant clustering was observed at <300 nm (*Figure 2—figure supplement 1F*).

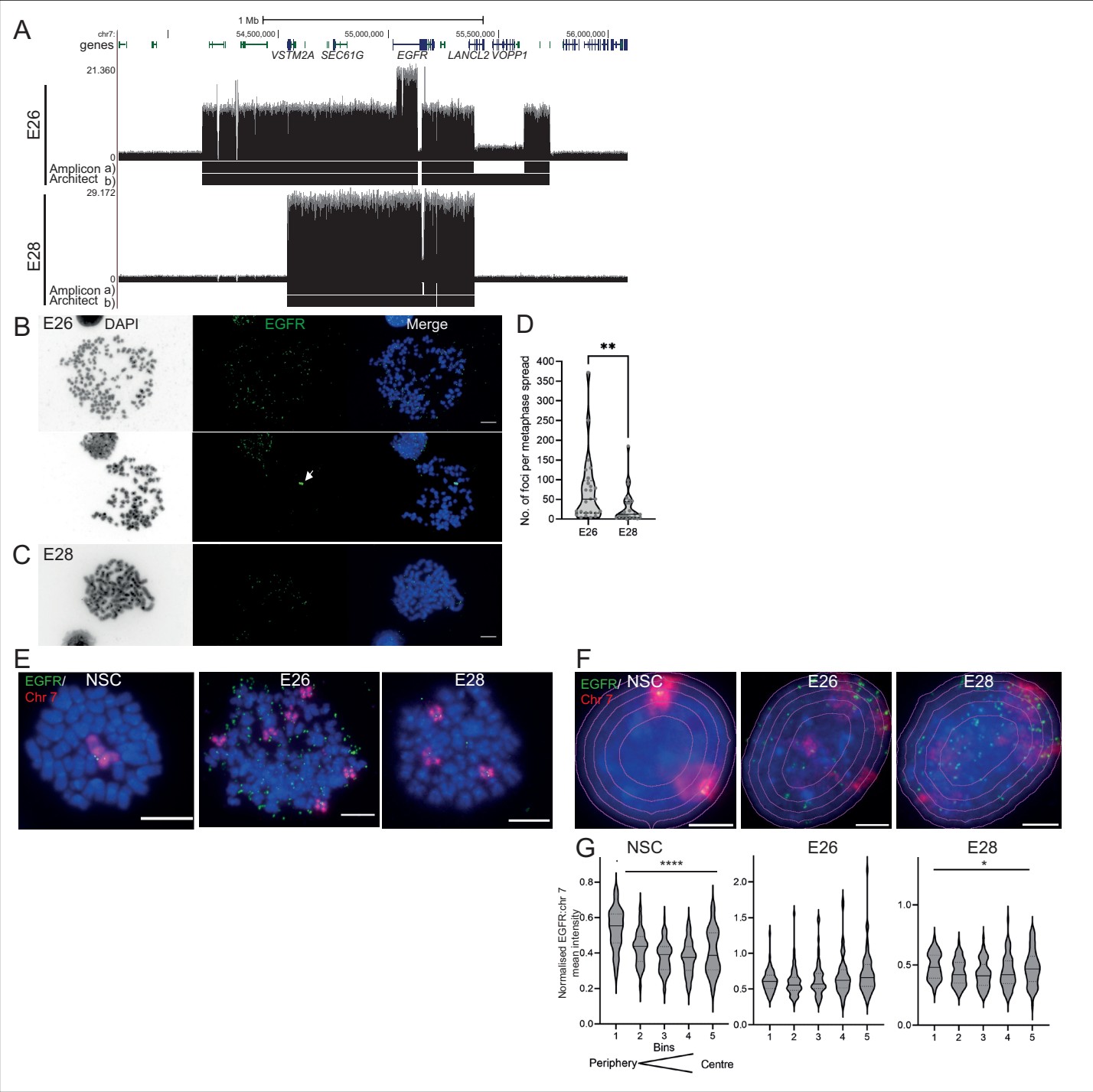

**Figure 1.** The nuclear localization of extrachromosomal DNA (ecDNA) in glioblastoma (GBM) cell lines. (**A**) Whole genome sequencing (WGS) and AmpliconArchitect analysis for ecDNA regions for E26 and E28 cell lines showing an *EGFR* exon 2–7 deletion in all ecDNA in E26 cells (seen in WGS and AmpliconArchitect regions a and b), and a subpopulation of ecDNA in E28 with a deletion across *EGFR* exons 7–14 (seen in WGS and Amplicon Architect region a – no deletion in E28 AmpliconArchitect region b). Genome coordinates (bp) are from the hg38 assembly of the human genome. (**B**) DNA FISH on metaphase spread of the E26 cell line showing *EGFR* (green) present on ecDNA, and on a homogeneously staining region (HSR) (arrowed) detected in ~10% of metaphases. Scale bar = 10 μm. (**C**) As for (**B**) but for the E28 cell line. (**D**) Violin plot of the number of *EGFR* DNA FISH signals per metaphase spread of E26 and E28 cells. Median and quartiles are shown. ** p=0.008 (Mann-Whitney test). Median values are 51 (E26)and 12 (E28), n=25 (E26) and 24 (E28) spreads. (**E and F**) Representative DNA FISH images of metaphase spread (**E**) and 2D nuclei (**F**) for neural stem cell (NSC), E26, and E28 cells showing signals for chromosome 7 (red) and EGFR (green). Blue = DNA (DAPI). Scale bar = 10 μm. The five erosions bins from the periphery to the centre of the nucleus are shown in F. (**G**) *EGFR* FISH signal intensity normalized to that for chromosome 7 (EGFR:Chr7 Mean Intensity)

*Figure 1 continued on next page*

*Figure 1 continued*

across five bins of equal area eroded from the peripheral (Bin 1) to the centre (Bin 5) of the nucleus for NSC, E26, and E28 cell lines. Median and quartiles shown. **** p<0.0001, * p<0.05. Kruskall-Wallis test. EGFR and chr7 signal normalized to DAPI shown in *Figure 1—figure supplement 1*. n=66 (NSC), 59 (E26), 64 (E28) nuclei. Statistical data relevant for this figure are in *Figure 1—source data 1*.

The online version of this article includes the following source data and figure supplement(s) for figure 1:

Source data 1. Statistical data for *Figure 1* and *Figure 1—figure supplement 1*.

Figure supplement 1. Additional *EGFR* and chromosome 7 signal intensity data.

## Different ecDNA populations do not cluster in the nucleus of GBM stem cells

To ensure that multiple ecDNAs are not so tightly clustered that they cannot be resolved by FISH, we analysed another primary GBM cell line (E25) which has two different oncogenes carried on separate ecDNA populations: *CDK4* and *PDGFRA* (*Figure 3—figure supplement 1A, B*). There was no obvious clustering of the two ecDNA populations in the nuclei of E25 cells (*Figure 3A*). The relative frequency of *CDK4-CDK4*, *PDGFRA-PDGFRA,* and *CDK4-PDGFRA* distances of ≤200 nm was low (*Figure 3B*). Indeed, the mean shortest interprobe distances per nucleus were overwhelmingly >1 μm, suggesting ecDNA were generally not in close proximity (*Figure 3—figure supplement 1C*). The shortest inter-probe distances for *CDK4-CDK4* and *CDK4-PDGFRA* were shorter than for *PDGFRA-PDGFRA* foci, as expected given the higher copy number of *CDK4* ecDNA (*Figure 3—figure supplement 1B*); however, there was no significant difference in the shortest distance between *CDK4-CDK4* and *CDK4-PDGFRA* foci (*Figure 3—figure supplement 1D*). No two *CDK4* or two *PDGFRA* foci were <200 nm apart, and only four *CDK4-PDGFRA* distances were <200 nm (4/1011 [0.39%] *CDK4* foci, 4/518 [0.77%] *PDGFRA* foci) (*Figure 3—figure supplement 1D*). These data suggest that clustering is not a significant feature of two separate populations of ecDNA.

We used 3D Ripley's K function to evaluate point patterns in the E25 dual ecDNA oncogene cell line (*Figure 3C*). Some nuclei had a significant non-random distribution of *PDGFRA* ecDNA at ≥400 nm, and most nuclei had non-random distribution of *CDK4* ecDNA at >400 nm (*Figure 3D*). When both foci were combined, there was no significant clustering at <300 nm in any nucleus, and the number of nuclei with a significant non-random distribution at a given radius rose with increasing radial distance (*Figure 3D*). As previously, a repeat analysis with a smaller (150 nm diameter) spot size identified no instances of significant clustering at <300 nm (*Figure 3—figure supplement 1E*).

To further validate this, we repeated 3D Ripley's function analysis in a second GBM cell line (E20) harbouring *CDK4* and *PDGFRA* ecDNAs. Whilst in the majority of metaphase spreads these two onco-genes were on clearly separate ecDNAs, in approximately 10% of metaphase spreads we noted colo-calization of *CDK4* and *PDGFRA* hybridization signals indicating a subset of ecDNA harbouring both oncogenes (*Figure 3—figure supplement 2A, B*). This colocalization could be observed in a similar proportion of interphase nuclei (*Figure 3E and F*). However, as observed in E25 cells the relative frequency of *CDK4-CDK4*, *PDGFRA-PDGFRA,* and *CDK4-PDGFRA* distances of ≤200 nm was low in the nucleus of E20 cells (*Figure 3H*). Ripley's K function analysis of hybridization signals in most E20 nuclei (22/24) showed no evidence for significant clustering of *CDK4* or *PDGFRA* at <300 nm (*Figure 3I*). We noted 2/24 (8.3%) of interphase nuclei (e.g. *Figure 3F*, see inset) where Ripley's K function indicated clustering of *CDK4* and *PDGFRA* foci at 100–200 nm and we suggest that these represent cells, as seen at metaphase, where the two oncogenes are located on the same ecDNA molecule. Doublets of *CDK4* foci (200 nm) were detected in 4/24 (16.7%) nuclei (*Figure 3G*, see inset).

Our analysis of two independent GBM cell lines harbouring different ecDNA populations (*CDK4* and *PDGFRA*) provides no evidence for systematic clustering of ecDNA molecules in the nucleus at distances <200 nm.

## ecDNA do not colocalize with large RNA PolII hubs in GBM stem cells

DNA FISH detects all ecDNA, so it might be that only transcriptionally active elements cluster. There-fore, we used RNA FISH to detect nascent *EGFR* transcripts in the nuclei of GBM cells. As expected, nascent RNA FISH foci were more frequent in the *EGFR* ecDNA-harbouring cell lines than in NSCs and were more frequent in the E26 GBM cell line than in E28 (*Figure 4—figure supplement 1A and B*). As for DNA FISH, we found no evidence of clustering of sites of *EGFR* nascent transcription at <400 nm in

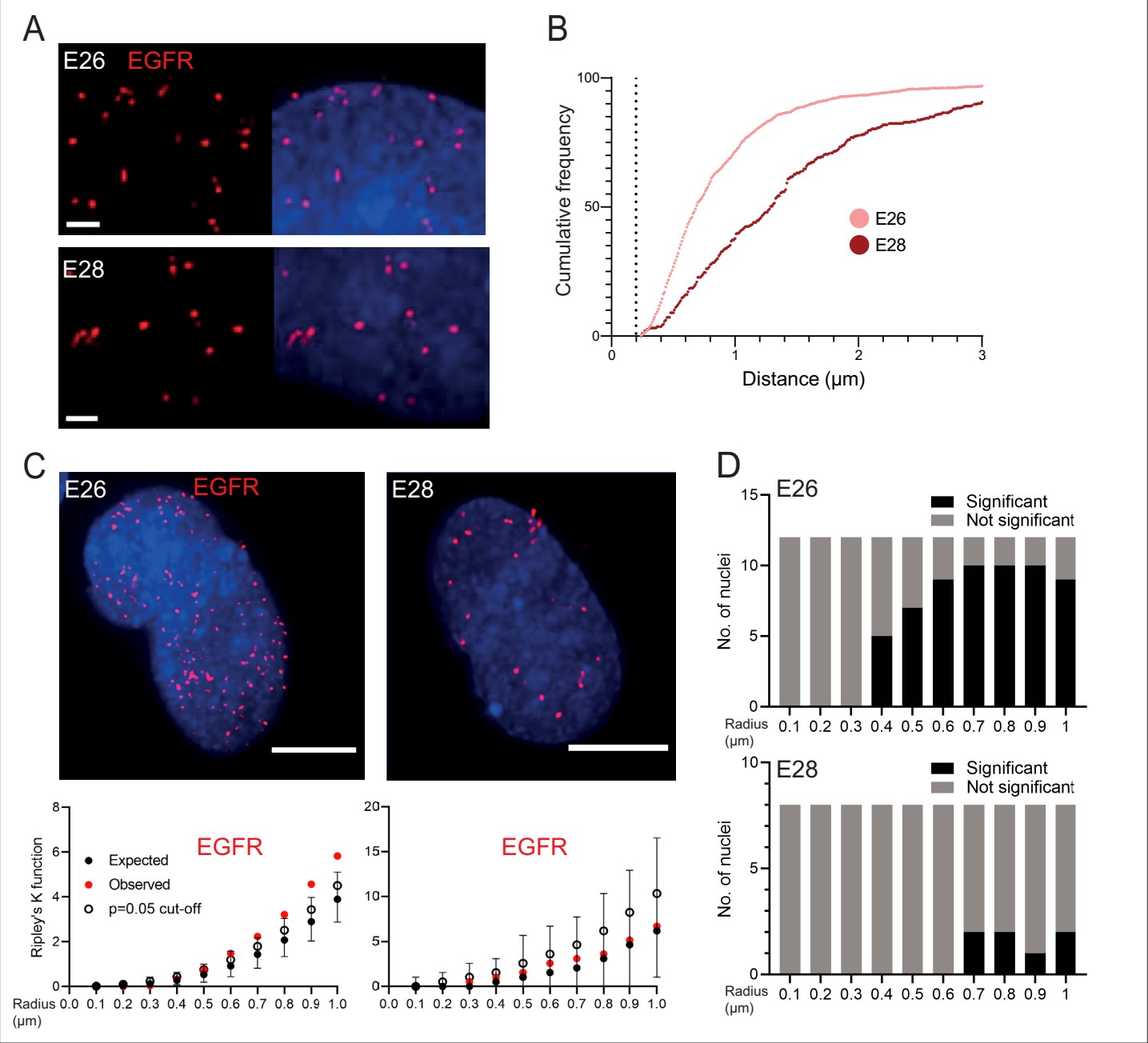

**Figure 2.** EGFR-containing extrachromosomal DNA (ecDNA) do not cluster in the nucleus. (**A**) Representative images shown as maximum intensity projection of DNA FISH for *EGFR* (red) in the nuclei of E26 (top) and E28 (bottom) glioblastoma (GBM) cell lines, scale bar = 1 µm. (**B**) Cumulative frequency distribution of shortest *EGFR-EGFR* distances between all foci in each nucleus across all E26 (n=37) and E28 (n=36) nuclei. Dotted line = 200nm. (**C**) (Top) Representative maximum intensity projection images of *EGFR* DNA FISH (red) in nuclei of E26 and E28 cells (blue=DNA). Scale bar = 5 µm. (Bottom) Associated 3D Ripley's K function for these nuclei showing observed K function (red), max/min/median (black) of 10,000 null samples with p=0.05 significance cut-off shown (empty black circle). (**D**) Ripley's K function for *EGFR* DNA FISH signals showing number of E26 (n=12) and E28 (n=8) nuclei with significant and non-significant clustering at each given radius. p-values were calculated using Neyman-Pearson lemma with optimistic estimate p-value where required (see Materials and methods), and Benjamini-Hochberg procedure (BHP, FDR = 0.05).

The online version of this article includes the following source data and figure supplement(s) for figure 2:

**Source data 1.** Statistical data for *Figure 2—figure supplement 1*.

**Figure supplement 1.** Additional analysis of *EGFR-EGFR* distances in E26 and E28 cell lines.

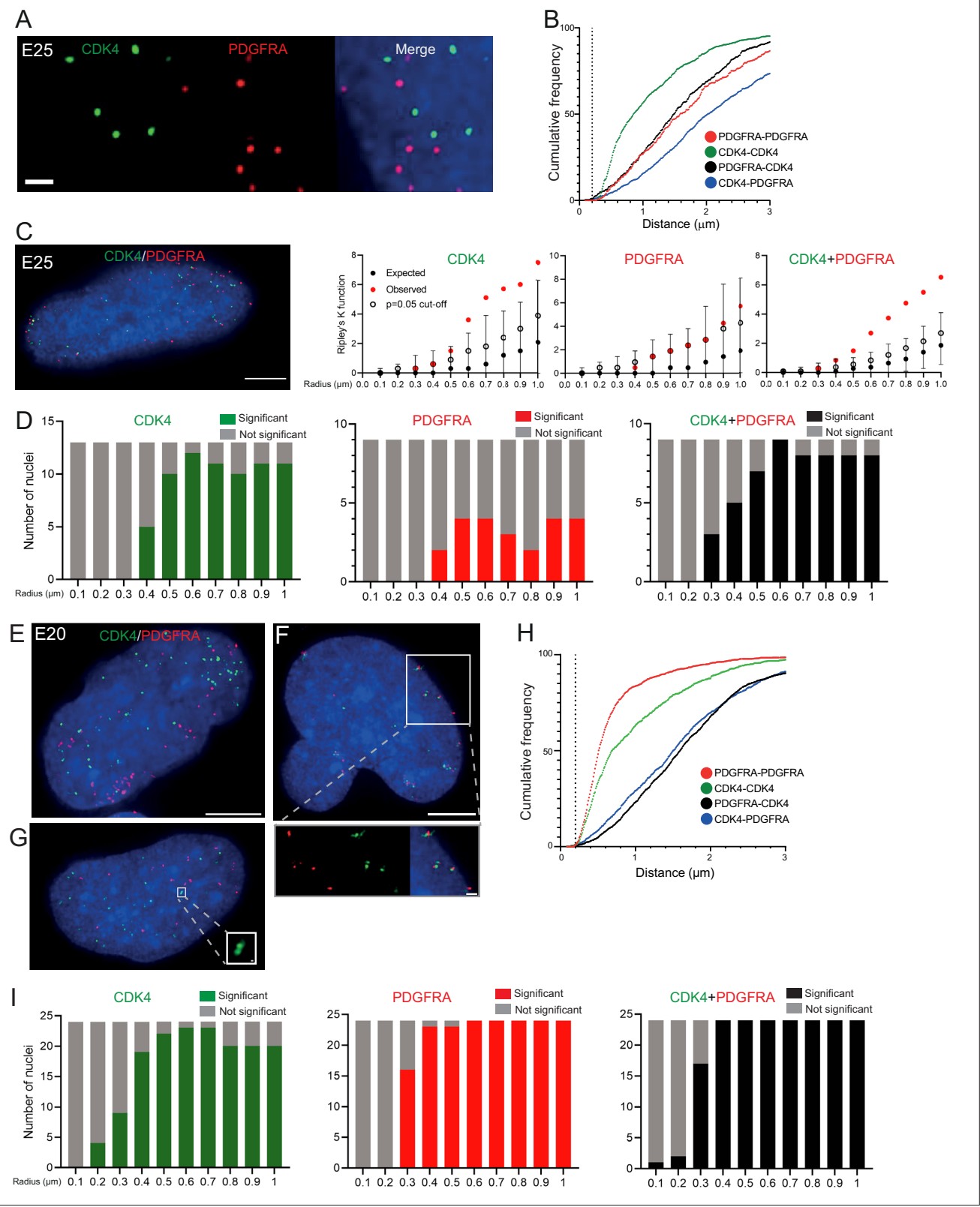

**Figure 3.** Two separate extrachromosomal DNA (ecDNA) populations do not cluster in the nucleus. (**A**) Representative maximum intensity projection images of DNA FISH for *CDK4* (green) and *PDGFRA* (red) in an E25 nucleus. Blue=DNA (DAPI) . Scale bar = 1 μm. (**B**) Cumulative frequency distribution of shortest interprobe distances (*CDK-CDK, PDGFRA-PDGFRA, CDK4-PDGFRA,* and *PDGFRA-CDK4*) between all foci in each nucleus across all E25 nuclei (n=26). (**C**) (Left) Representative maximum intensity projection image shown of E25 nucleus hybridized with probes for *CDK4* (green) and

*Figure 3 continued on next page*

*Figure 3 continued*

*PDGFRA* (red). Blue=DNA (DAPI). Scale bar = 5 µm. (Right) Ripley's K function for this nucleus showing observed K function (red), max/min/median (black) of 10,000 null samples with p=0.05 significance cut-off shown (empty black circle) for CDK4, *PDGFRA*, and *CDK4* and *PDGFRA* spots combined. (**D**) Ripley's K function for E25 nuclei showing number of nuclei with significant and non-significant clustering at each given radius for *CDK4* spots (n=13 nuclei), *PDGFRA* spots (n=9 nuclei), and *CDK4* and *PDGFRA* spots combined (n=9 nuclei). p-values were calculated using Neyman-Pearson lemma with optimistic estimate p-value where required (see Materials and methods), and Benjamini-Hochberg procedure (BHP, FDR = 0.05). Metaphase analysis of E25 cells and Ripley's K analysis with smaller foci are in *Figure 3—figure supplement 1*. (**E**) Representative maximum intensity projection image of E20 interphase nuclei hybridized with probes for *CDK4* (green) and *PDGFRA* (red). Scale bar = 5 µm. (**F**) As in (**E**) but for a nucleus where the close association of CDK4 and PDGFA signal in doublets is indicative of ecDNAs harbouring both oncogenes. Scale bar = 1 µm in main panel (**G**) as in (**E**) but showing an E20 nucleus with doublets of CDK4 foci. Metaphase analysis of E20 cells with *CDK4* and *PDGFRA* probes in *Figure 3—figure supplement 2*. (**H**) As in (**B**) but for E20 nuclei (n=24) (noting all nuclei shown here harbored >20 foci of each oncogene). (**I**) As in (**D**) but for E20 nuclei.

The online version of this article includes the following source data and figure supplement(s) for figure 3:

**Source data 1.** Statistical data for *Figure 3—figure supplement 1*.

**Figure supplement 1.** Additional analysis of the distribution of *CDK4* and *PDGFRA* ecDNAs in the E25 cell line.

**Figure supplement 2.** DNA FISH on metaphase spreads of the E20 cell line showing hybridization signal for *PDGFRA* (red) and *CDK4* (green).

---

E26 cells (*Figure 4A and B*). These data suggest that ecDNA actively transcribing a driver oncogene do not colocalize in the nucleus of GBM cells more than expected by chance.

We next assessed whether ecDNA foci, albeit not clustered with each other, colocalize with high focal concentrations of the transcriptional machinery to create ecDNA/large PolII transcription hubs. First, we examined the presence of such hubs by immunofluorescence for RPB1 (POLR2A), the largest subunit of RNA PolII. The large RPB1 foci we detected were sparse with only a few clearly visible per nucleus (*Figure 4—figure supplement 1C*).

We used 3D analysis of immunoFISH in NSCs and compared this to E26 and E28 GBM cells to establish whether ecDNA and large RPB1 foci colocalized. There was no obvious overlap between foci of RPB1 and *EGFR* (*Figure 4C*) and no correlation between the number of large RPB1 foci and the number of *EGFR* foci (*Figure 4D*). Indeed, the mean shortest distance between *EGFR* foci and large RPB1 foci per nucleus was routinely >1 µm in all cell lines, despite the greater number of *EGFR* foci in the GBM cell lines (*Figure 4E*). The single shortest distance per nucleus between an *EGFR* locus and a large RPB1 locus was not significantly different across NSC and tumour lines (*Figure 4F*). There were no instances where the distance between *EGFR* and large RPB1 foci was <200 nm. To test if this was also the case for the nascent *EGFR* RNA transcript, we repeated this analysis using nascent RNA FISH, with the same result (*Figure 4—figure supplement 1D–F*). As the distance distributions to large RPB1 foci were similar for DNA and RNA FISH, this suggests that proximity to large PolII hubs does not alter the probability that ecDNA are transcribed.

To ensure this result was not specific to this PolII antibody, we repeated this analysis using E28 cells in which mCherry was fused by knock-in to endogenous POLR2G, a key subunit of RNA PolII (*Cramer et al., 2000*). The mean distance between *EGFR* foci and large POLR2G foci and the shortest minimum distance in any given nucleus (*Figure 4—figure supplement 1G–I*) further support that there is no close spatial relationship apparent between ecDNA and large PolII hubs.

## Levels of EGFR transcription from ecDNA reflect copy number, not enhanced transcriptional efficiency

Having shown a lack of colocalization of ecDNA, either with each other or with large PolII foci, we proceeded to characterize the levels of *EGFR* expression from ecDNA. Flow cytometry using a fluorophore-conjugated EGFR ligand (EGF-647) revealed consistently higher levels of EGFR in the GBM cells than NSC, with highest signal in E26 (*Figure 5—figure supplement 1A, B*), consistent with their higher ecDNA copy number compared with E28 (*Figure 1C*). To confirm this link between ecDNA number and levels of EGFR, E26 and E28 cells were sorted by fluorescence activated cell sorting (FACS) into EGFR-high and EGFR-low populations. In both tumour cell lines, DNA FISH demonstrated that EGFR-high cells had a significantly higher number of *EGFR* DNA foci than EGFR-low (*Figure 5—figure supplement 1C–E*).

Previous studies have reported that ecDNA have greater transcript production per oncogene than chromosomal loci (*Wu et al., 2019*). We therefore sought to characterize the transcriptional efficiency (per copy number) of chromosomal and ecDNA-located *EGFR* genes in our GBM cell lines, by

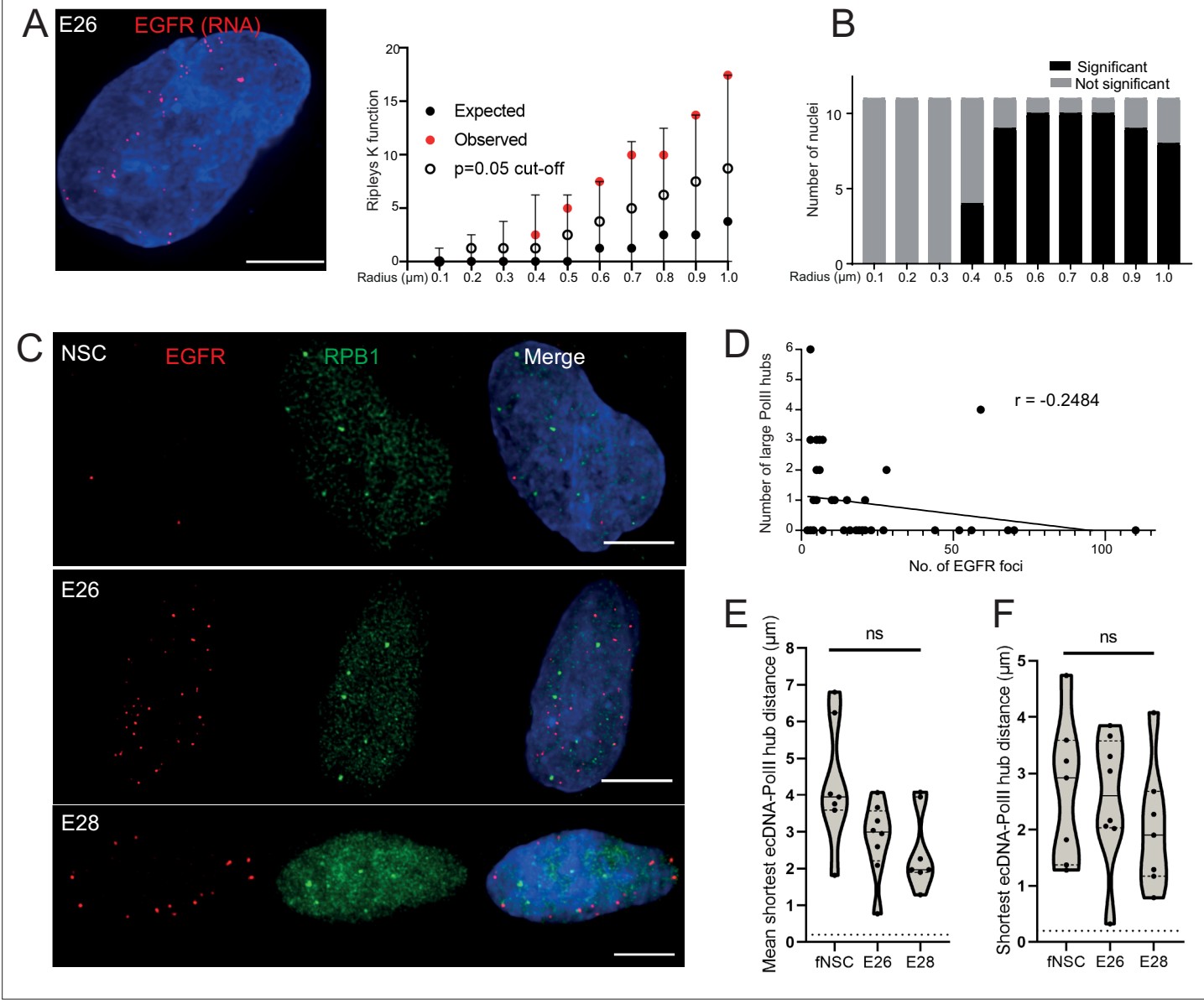

**Figure 4.** Extrachromosomal DNA (ecDNA) do not colocalize with large foci of the transcriptional machinery. (**A**) Representative maximum intensity projection image of nascent *EGFR* RNA FISH (red) in E26 cell nucleus,(blue=DNA). Scale bar = 5 µm. Associated Ripley's K function for this nucleus showing observed K function (red), max/min/median (black) of 10,000 null samples with p=0.05 significance cut-off shown (empty black circle). (**B**) Ripley's K function for E26 nuclei (n=11) after *EGFR* nascent RNA FISH showing number of nuclei with significant and non-significant clustering at each given radius. All p-values for Ripley's K function calculated using Neyman-Pearson lemma with optimistic estimate p-value where required, and Benjamini-Hochberg procedure (BHP, FDR = 0.05). (**C**) Representative maximum intensity projection images of immunoFISH in neural stem cell (NSC), E26 and E28 cell lines: Immunofluorescence for RPB1 (green) and *EGFR* DNA FISH (red). Scale bar = 5 µm. (**D**) Spearman's correlation between number of *EGFR* foci and number of RPB1 foci, p = 0.13, E26 and E28 cell line data combined. (**E**) Violin plot of distribution of mean shortest interprobe distance per nucleus between *EGFR* foci and PolII foci in NSC (n=7), E26 (n=8) and E28 (n=7) cell lines. (**F**) As for (**E**) but for shortest single distance in each nucleus. ns, not significant. Kruskall-Wallis test. Statistical data relevant for this figure are in *Figure 4—source data 1*.

The online version of this article includes the following source data and figure supplement(s) for figure 4:

**Source data 1.** Statistical data for *Figure 4* and *Figure 4—figure supplement 1*.

**Figure supplement 1.** Analysis of sites of EGFR nascent transcription relative to RNA polymerase II in GBM cell lines.

assaying the RNA:DNA *EGFR* FISH foci ratio. We performed nascent *EGFR* RNA FISH using a probe targeting the first intron of *EGFR* and *EGFR* DNA FISH to test this hypothesis (*Figure 5A*).

When comparing the RNA:DNA ratio of all nuclei, only E26 had a higher ratio than NSCs (*Figure 5B*). To explore whether *EGFR* transcription in these cell lines could be due to ec*EGFR*-driven increased transcriptional efficiency, we used chromosome 7 copy number (evaluated by CEN7 probe) to account for chromosomal *EGFR* copy number. We correlated the RNA:DNA FISH ratio with the proportion of ec*EGFR* (number of *EGFR* foci minus number of CEN7 foci, divided by the total number of EGFR foci). We observed no correlation in either cell line (*Figure 5C*), suggesting that *EGFR* transcription from ecDNA and chromosomes occurs at similar levels when normalized to chromosome 7 copy number. There is no increased transcriptional efficiency from ecDNA compared to chromosomal DNA based on these analyses.

To test this using an independent method, we took advantage of WGS and RNA-seq data (*Figure 5D*) and called SNPs present in the amplicon region at 40% to 60% allele frequencies in patient control blood WGS (control) samples. Most of the allele frequencies of these SNPs were >80% in GBM samples in the main part of the amplicons, in line with the amplification being derived from one parental allele (*Figure 5—figure supplement 1F*). We then selected those SNPs located in expressed exons of the amplicon, including several in *EGFR*. The WGS allele frequencies of these were all >88%, that is, predominantly from amplicons. If genes on the ecDNA are more highly transcribed than chromosomal counterparts, we expect the ratio of RNA-seq to WGS reads of the amplicon-derived SNP to be above 1. Consistent with genes on ecDNA and on chromosomes being transcribed with similar efficiencies, these values were very close to 1, the highest being 1.05 (*Figure 5D, E*). The lower values for *LANCL2*, 3′ of *EGFR*, are likely because only part of this gene is present on the amplicon such that the transcript is truncated. As an additional approach, we utilized the large exon 2–7 deletion present on E26 *EGFR* ecDNA to compare the copy number-normalized RNA expression of exons present only on the endogenous chromosomal *EGFR* locus (exons 2–7) with those predominantly on ecDNA (exons 1, 8–28) (*Figure 5E, D*). Copy number normalized *EGFR* RNA counts were not significantly different between exons 2–7 and those located predominantly on ecDNA (*Figure 5F*). EcDNA with *EGFR* in another established GBM cell line, GBM39, also contain a deletion spanning exons 2–7. We therefore repeated this analysis using previously published WGS and RNA-seq data from this cell line (*Wu et al., 2019*). The normalized RNA read count of primarily ec*EGFR* exons was not significantly different than that of chromosomal *EGFR* exons (*Figure 5G*). Altogether, RNA:DNA FISH and sequencing analyses suggest that *EGFR* on each ecDNA is transcribed at a similar level to that of the corresponding endogenous chromosomal *EGFR* locus. Increased output of oncogenes in GBM stem cells with ecDNA appears to be primarily driven by increased copy number, rather than inherent features of their chromatin state, transcriptional control, or spatial localization.

## Discussion

Understanding the importance of ecDNA in the etiology of cancer, and whether this poses an interesting target for therapeutic interventions, depends on deeper analysis of ecDNA activity (*Nathanson et al., 2014*; *Kim et al., 2020*). Clustering of ecDNA into 'ecDNA hubs' based on imaging and chromosome conformation capture data has been reported in a range of established cancer cell lines, and has been suggested to underlie the ability of ecDNA to drive very high levels of transcription (*Hung et al., 2021*; *Yi et al., 2021*; *Zhu et al., 2021*). However, in multiple primary human GBM cells studied here, we observe no significant colocalization at distances (~200 nm) thought to be functionally important in driving transcription. We reach this conclusion for both cells with single ecDNA species, as well as with heterogeneous ecDNA harbouring different oncogenes. EcDNA were not colocalized with, or notably close to, large PolII foci. Moreover, taking advantage of the unique transcripts from ecDNA, and the presence of SNPs in these transcripts, to compare ecDNA-derived and chromosomal transcripts, we demonstrate that increased copy number primarily drives increased transcription of ecDNA-located genes rather than increased transcriptional efficiency of ecDNA in GBM stem cells.

Our data support a regional, rather than clustered, spatial organization of ecDNA in GBM stem cells. We observe that oncogenes on ecDNA are distributed more towards the centre of the nucleus than the corresponding endogenous gene loci. This is consistent with an actively transcribing state (*Boyle et al., 2001*; *Croft et al., 1999*) and independence from the constraints of chromosome territories (*Kalhor et al., 2011*; *Mahy et al., 2002*).

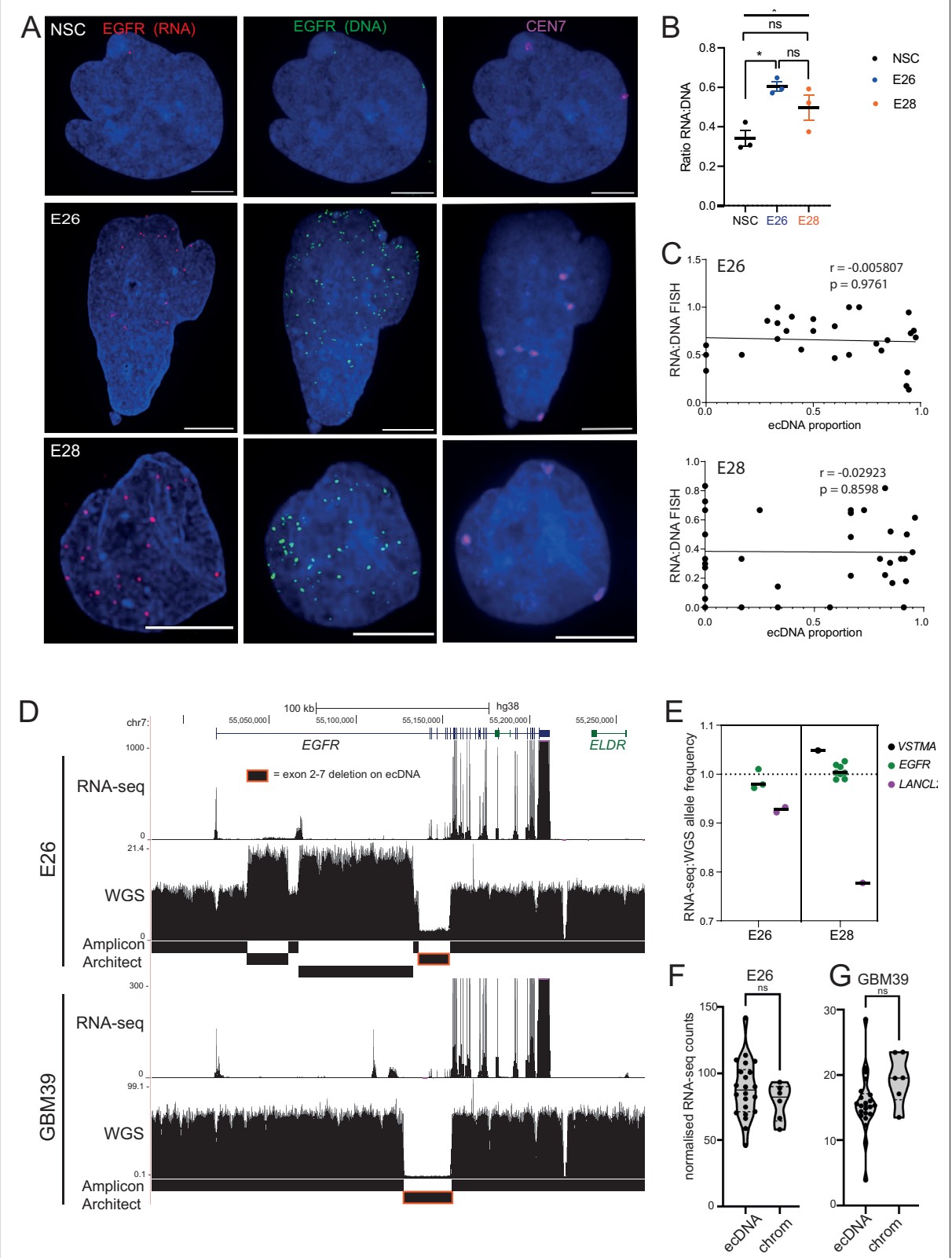

**Figure 5.** Levels of transcription from extrachromosomal DNA (ecDNA) reflect copy number but not enhanced transcriptional efficiency. (**A**) Representative maximum intensity projection (MIP) images of nascent *EGFR* RNA, *EGFR,* and centromere 7 (CEN7) DNA FISH in neural stem cell (NSC), E26 and E28 cell lines (scale bar = 5 µm). (**B**) Ratio of RNA:DNA foci per nucleus in NSC, E26 and E28 cell lines. * p<0.05, n.s. not significant. Flat line – one-way ANOVA, hooked lines – unpaired t-test. Mean and standard error of the mean (SEM) plotted, with 3 biological replicates for NSC (total

*Figure 5 continued on next page*

*Figure 5 continued*

n=67), E26 (98) and E28 (95) nuclei. (**C**) Representative Spearman r correlation ($\rho$) and p-values shown for E26 (n=29) and E28 (n=39) cells. RNA:DNA ratio = number of RNA foci/number of DNA foci. EcDNA proportion = (number of *EGFR* DNA foci – number of CEN7 foci)/number of *EGFR* DNA foci. Three biological replicates performed, data from replicate 1 shown here. (**D**) UCSC genome browser tracks showing E26 and GBM39 RNA-seq and WGS aligned sequences in the region of chromosome 7 where *EGFR* is located, *EGFR* exons (GENCODE) and the exon deletion predicted by AmpliconArchitect. Note that RNA-seq counts in some ecDNA regions go above the maximum value. Genome coordinates (Mb) are from the hg38 assembly of the human genome. (**E**) RNA-seq/WGS allele frequency ratio for SNPs overlapping with expressed exons in the amplicon. Lines denote median values. (**F**) *EGFR* RNA-seq counts normalized by WGS read count per *EGFR* exon in E26, with exons defined as extrachromosomal (exons 1,8-28) or chromosomal (exons 2-7). Statistical significance examined by Mann-Whitney test. ns, not significant. (**G**) As for (**F**) but for GBM39. Statistical data relevant for this figure are in ***Figure 5—source data 1***.

The online version of this article includes the following source data and figure supplement(s) for figure 5:

**Source data 1.** Statistical data for ***Figure 5*** and ***Figure 5—figure supplement 1***.

**Figure supplement 1.** EGFR levels, ecDNA number, and ecDNA SNP allele frequency in E26 and E28 cell lines.

We sought to maximize our opportunity of observing ecDNA clustering at close distances by performing 3D spot analysis, using Ripley's K to call instances of significant clustering at given distances using ecDNA x,y,z coordinates, and utilizing cells with two distinct ecDNA species to ensure we were not under-scoring colocalization. 3D analysis ensures a false positive clustering effect is avoided that might be seen when 3D images are combined via tools such as maximum intensity projection (MIP). Other tools to assess clustering have noted the possibility of the 2D Ripley's K function resulting in over-counting, leading to the development of alternative auto-correlation tools, but this was not observed in this 3D Ripley's K analysis (***Veatch et al., 2012***). It is possible that multiple clustered DNA/RNA foci appear as a single DNA/RNA FISH signal that we cannot resolve. We controlled for this by repeating cluster analysis with smaller spot sizes, analyzing cell lines with two ecDNA populations and using super-resolution imaging (optical resolution ~120 nm). We did observe ecDNA clustering at close distances (≤200 nm) in a small proportion of E20 dual-ecDNA cells, but in the case of *CDK4-PDGFRA* colocalization this was at a similar proportion to that observed in metaphase spreads, indicative of ecDNA molecules harbouring both *CDK* and *PDGFRA*. The incidence of *CDK4* doublets (which appeared in keeping with double minutes) was also low. Overall, this suggests that close clustering is not a major contributor to increased ecDNA transcriptional output in GBM stem cells.

Our findings may reflect fundamentally different functional characteristics of the ecDNA in patient-derived primary GBM cell cultures used in our experiments versus previously published studies (***Hung et al., 2021***; ***Yi et al., 2021***). These might include the size of the ecDNA, or the number of oncogene loci per ecDNA (which was singular in our cell lines, with the exception of ~10% E20 *CDK4/PDGFRA* colocalized ecDNA). For example, the COLO320-DM cell line, used in a recent study of ecDNA hubs, harbours 3 copies of *MYC* on each of its ecDNA, and results in large (4.328 Mb, approx. 1.75 µm diameter) ecDNA (***Hung et al., 2021***; ***Wu et al., 2019***). The HK359 GBM cell line, previously noted to have clustered ecDNA hubs, has a 42 kb insertion at the site of *EGFR*vIII (exon 2–7 deletion), again suggesting a large ecDNA quite different in character to those described here (***Hung et al., 2021***; ***Koga et al., 2018***). More quantitative analysis across a larger set of primary cancer cells will be needed to determine if long-term established cell lines have unusual ecDNA features and are unrepresentative of primary GBM cells.

Recent work proposing that ecDNA act as mobile super-enhancers for chromosomal targets has raised the possibility that ecDNA can actively recruit RNA PolII to drive 'ecDNA-associated phase separation' (***Zhu et al., 2021***). A live-cell ecDNA-labelling strategy reported colocalization of ecDNA and RNA PolII (***Yi et al., 2021***). We did not detect evidence of a close relationship between ecDNA, or their nascent transcript, with large PolII foci, but cannot exclude that there are smaller, sub-diffraction limit sized transcriptional hubs associated with our ecDNA.

We observe that while the copy number of *EGFR* ecDNAs positively correlates with greater transcriptional output, this is likely due to copy number increases, rather than increased transcriptional activity on individual ecDNA. It has been proposed that ecDNA increase transcription of their resident oncogenes partly due to their increased DNA copy number, but also due to their more accessible chromatin structure, and that gene transcription from circular amplicons is greater than that of linear amplicons once copy number normalized (***Kim et al., 2020***; ***Wu et al., 2019***). An analysis of RNA-seq and WGS data from a cohort of 36 independent clinical samples found that only 3 out of

11 ecDNA-encoded genes produced significantly more transcripts when normalized to gene copy number, only one of which is a key oncogene (**Wu et al., 2019**). In agreement with this, our analysis of both oncogene and amplicon-resident polymorphisms suggests that copy number is the dominant driver of ecDNA gene transcription.

Overall, our data suggest that in primary GBM stem cells, ecDNA can succeed at driving oncogene expression without requiring close colocalization with each other, or with transcriptional hubs. It is the increased copy number that is primarily responsible for higher levels, rather than ecDNA-intrinsic features or nuclear sub-localization.

## Materials and methods

### Key resources table

| Reagent type (species) or resource | Designation | Source or reference | Identifiers | Additional information |
|---|---|---|---|---|
| Antibody | mCherry (Rabbit poly-clonal) | abcam | ab167453 | IF (1 in 500) |
| Antibody | Rpb1 NTD (D8L4Y) (Rabbit mono-clonal) | Cell Signaling Technology | #14958 | IF (1 in 1000) |
| Antibody | Anti-Digoxigenin (Sheep poly-clonal) | Roche | Ref 11333089001 | DNA FISH (1 in 10) |
| Antibody | Secondary Antibody – Alexa Fluor 647 (Donkey anti-Sheep IgG poly-clonal) | Thermo Fisher Scientific | A-21448 | DNA FISH (1 in 10) |
| Antibody | Secondary Antibody – Alexa Fluor 568 (Donkey anti-Rabbit IgG poly-clonal) | Thermo Fisher Scientific | A-10042 | IF (1 in 1000) |
| Antibody | Secondary Antibody – Alexa Fluor 488 (Donkey anti-Rabbit IgG poly-clonal) | Thermo Fisher Scientific | A-21206 | IF (1 in 1000) |
| Antibody | Secondary Antibody – Alexa Fluor 488 (Donkey anti-Rat IgG poly-clonal) | Thermo Fisher Scientific | A-21208 | IF (1 in 1000) |
| Genetic reagent (human) | Fosmid FISH probe (Human) | BACPAC resource | https://bacpacresources.org/library.php?id=275 | See Materials and methods - **Supplementary file 1** |
| Cell line (*Homo sapiens*) | E20, E25, E26, E28, NSC – GCGR Human Glioma Stem Cells | This paper, Glioma Cellular Genetics Resource, CRUK, UK | http://gcgr.org.uk; pending publication | |
| Other | DMEM/HAMS-F12 | Sigma-Aldrich | Cat#: D8437 | Cell culture, media |
| Chemical compound, drug | Pen/Strep | GIBCO | Cat#: 15140–122 | Cell culture, media supplement |
| Other | BSA Solution | GIBCO | Cat#: 15260–037 | Cell culture, media supplement |
| Other | B27 Supplement (×50) | LifeTech/GIBCO | Cat#: 17504–044 | Cell culture, media supplement |
| Other | N2 Supplement (×100) | LifeTech/GIBCO | Cat#: 17502–048 | Cell culture, media supplement |
| Other | Laminin | Cultrex | Cat#: 3446-005-01 | Cell culture, media supplement, and pre-lamination of culture vessels |
| Peptide, recombinant protein | EGF | Peprotech | Cat: 315–09 | Cell culture, media supplement |
| Peptide, recombinant protein | FGF-2 | Peprotech | 100-18B | Cell culture, media supplement |
| Other | Accutase | Sigma-Aldrich | Cat#: A6964 | Cell culture, cell dissociation agent |
| Other | DMSO | Sigma-Aldrich | Cat#: 276855 | Cell culture, freeze media, and drug diluent |

*Continued on next page*

*Continued*

| Reagent type (species) or resource | Designation | Source or reference | Identifiers | Additional information |
|---|---|---|---|---|
| Other | Triton X-100 | Merck Life Sciences | Cat#: X-100 | Cell permeabiliz-ation agent following cell fixation |
| Other | Paraformaldehyde Powder 95% | Sigma | Cat#: 158127 | Cell fixation agent |
| Other | Tween 20 | Cambridge Bioscience | Cat#: TW0020 | DNA FISH (hybridization mix) |
| Other | PBS Tablets | Sigma-Aldrich | Cat#: P4417 | Diluent and washing agent |
| Other | Ethanol | VWR | Cat#: 20821–330 | DNA FISH |
| Other | Methanol | Fisher Chemical | M/4000/17 | Used 3:1 with acetic acid for metaphase spreads |
| Other | Acetic acid | Honeywell Research Chemicals | 33209-1L | See above |
| Peptide, recombinant protein | Alexa Fluor 647 EGF complex | Thermo Fisher Scientific | E35351 | Flow cytometry |
| Other | Green496-dUTP | ENZO Life Sciences | ENZ-42831L | Direct labelling of Fosmid DNA FISH probes via nick translation |
| Other | ChromaTide Alexa Fluor 594–5-dUTP | Thermo Fisher Scientific | C11400 | Direct labelling of Fosmid DNA FISH probes via nick translation |
| Peptide, recombinant protein | DNA Polymerase 1 | Invitrogen | 18010–017 | |
| Peptide, recombinant protein | DNase I recombinant, RNase-free | Roche | 04716728001 | |
| Genetic reagent (human) | Human Cot-1 DNA | Thermo Fisher Scientific | 15279011 | |
| Genetic reagent (salmon) | Salmon Sperm DNA | Invitrogen | 15632011 | |
| Chemical compound, drug | Paclitaxel | Cambridge Bioscience | CAY10461 | 10–100 nM |
| Chemical compound, drug | Nocodazole | Sigma-Aldrich | SML1665 | 50–100 ng/ml |
| Other | XCP 7 Orange Chromosome Paint | MetaSystems Probes | D-0307-100-OR | DNA FISH (see *Figure 1* and Materials and methods referring to this) |
| Commercial assay or kit | Stellaris RNA-FISH probes (Custom Assay with Quasar 570 Dye) | LGC Biosearch Technologies | SMF-1063–5 | RNA FISH |
| Commercial assay or kit | Stellaris RNA FISH Hybridization Buffer | LGC Biosearch Technologies | SMF-HB1-10 | RNA FISH |
| Genetic reagent (human) | Alt-R CRISPR-Cas9 crRNA | IDT-Technologies | Alt-R CRISPR-Cas9 crRNA | |
| Genetic reagent (human) | Alt-R CRISPR-Cas9 tracrRNA | IDT-Technologies | 1072532 | |
| Commercial assay or kit | SG Cell Line 4D-NucleofectorTM X Kit S | Lonza Bioscience | V4XC-3032 | |
| Genetic reagent (human) | Chromosome 7 Control Probe | Pisces Scientific | CHR07-10-DIG | Probe and hybridization mix |
| Other | DAPI (4',6-Diamidino-2-Phenylindole, Dihydrochloride) | Thermo Fisher Scientific | D1306 | Nuclear staining; 50 ng/ml and 5 ng/ml (as indicated in Materials and methods) |

*Continued on next page*

*Continued*

| Reagent type (species) or resource | Designation | Source or reference | Identifiers | Additional information |
|---|---|---|---|---|
| Sequence-based reagent | mCherry_PolR2G crRNA and dsDNA (donor) | Twist Bioscience | | See Materials and methods and ***Supplementary file 1*** |
| Other | WGS and RNAseq | This paper Glioma Cellular Genetics Resource, CRUK, UK | GEO: GSE215420 See also: https://gcgr.org.uk | See Materials and methods |
| Other | Erosion Territories analysis | This paper | | Code available at: https://github.com/IGC-Advanced-Imaging-Resource/Purshouse2022_paper |
| Other | Cluster analysis | This paper | | Code available at: https://github.com/SjoerdVBeentjes/ripleyk |
| Other | RNA-seq/WGS analysis | This paper | | Code available at: https://github.com/kpurshouse/ecDNAcluster |
| Software, algorithm | GraphPad Prism 9.0 | GraphPad Software, Inc | https://www.graphpad.com/ | |
| Software, algorithm | FCS Express | FCS Express 7 | https://denovosoftware.com/ | |
| Software, algorithm | Fiji/ImageJ | Open Source | https://imagej.net/Fiji | |
| Software, algorithm | BioRender | BioRender | https://biorender.com/ | |
| Software, algorithm | Python v3.9 | Open Source | https://www.python.org | |
| Software, algorithm | Algorithm - RipleyK package | Python Package Index | https://pypi.org/project/ripleyk/ | |
| Software, algorithm | Imaris x64 v9.4.0 | Imaris Microscopy Image Analysis Software | https://imaris.oxinst.com/ | |
| Software, algorithm | UCSC Genome Browser | ***Kent et al., 2002*** | https://genome.cshlp.org/content/12/6/996 | |
| Software, algorithm | STAR 2.7.1a | ***Dobin et al., 2013*** | https://github.com/alexdobin/STAR; ***Dobin et al., 2013*** | |
| Software, algorithm | Picard | Broad Institute | https://broadinstitute.github.io/picard/ RRID:SCR_006525, Version 2.23.2 | |
| Software, algorithm | AmpliconArchitect | ***Deshpande et al., 2019*** | https://github.com/virajbdeshpande/AmpliconArchitect; ***Deshpande et al., 2019*** (with Python v2.7) | |
| Software, algorithm | AmpliconClassifier | ***Kim et al., 2020*** | https://github.com/jluebeck/AmpliconClassifier (with Python v2.7) | |
| Software, algorithm | deepTools v3.4 | ***Ramírez et al., 2016*** | https://deeptools.readthedocs.io/en/develop/ | |
| Software, algorithm | HOMER2 4.10 | ***Heinz et al., 2010*** | http://homer.ucsd.edu/homer/ | |
| Software, algorithm | SAMtools v1.10 | ***Li et al., 2009*** | http://www.htslib.org | |
| Software, algorithm | BEDTools v2.3 | ***Quinlan and Hall, 2010*** | http://code.google.com/p/bedtools | |
| Software, algorithm | bcftools | ***Danecek et al., 2021*** | https://doi.org/10.1093/gigascience/giab008 | |
| Software, algorithm | strelka v2.9.10 | ***Kim et al., 2018*** | https://doi.org/10.1038/s41592-018-0051-x | |

## Lead contact

Further information and requests for resources and reagents should be directed to and will be fulfilled by the lead contacts, Wendy Bickmore (wendy.bickmore@ed.ac.uk) and Steven Pollard (steven.

pollard@ed.ac.uk).

## Materials availability

This study generated a new CRISPR engineered knock-in reporter cell line – E28 mCherry_POLR2G.

## Experimental model and subject details

GSC and NSC lines from the Glioma Cellular Genetics Resource (GCGR) (https://gcgr.org.uk) were cultured in serum-free basal DMEM/F12 medium (Sigma) supplemented with N2 and B27 (Life Technologies), 2 µg/ml laminin (Cultrex), and 10 ng/ml growth factors EGF and FGF-2 (Peprotech) (*Pollard et al., 2009*). Cells were split with Accutase solution (Sigma), and centrifuged approximately weekly as previously reported. All GBM cell lines were derived from treatment-naive patients, and the NSC cell line GCGR-NS9FB_B was derived from 9 week of gestation forebrain. GSC cell lines were selected on the basis of predominantly (E26) or entirely (E28, E25, and E20) harbouring oncogenes on ecDNAs (rather than HSRs) via metaphase spread analysis (see Materials and method below). Human GBM tissue was obtained with informed consent and ethical approval (East of Scotland Research Ethics service, REC reference 15/ES/0094). Human embryonic brain tissue was obtained with informed consent and ethical approval (South East Scotland Research Ethics Committee, REC reference 08/S1101/1). Cell lines were regularly tested for mycoplasma.

## Method details

### Metaphase spreads and interphase nuclei

Cell lines were optimized to generate metaphase spreads. Briefly, cells at near confluence in a T75 flask were incubated between 4 and 16 hr in the presence of 10–100 nm paclitaxel (Cambridge BioScience) with or without 50–100 ng/ml nocodazole (Sigma-Aldrich). Along with the media, cells dissociated with accutase were centrifuged, washed in PBS, and resuspended in 10 ml potassium chloride (KCl) 0.56%, with sodium citrate dihydrate (0.9%) if required, for 20 min. After further centrifugation, cells were resuspended in methanol:acetic acid 3:1 and dropped onto humidified slides.

For all other fixed cell experiments described below, cells were seeded overnight onto glass coverslips or poly-L-lysine coated glass slides (Sigma-Aldrich). Cells were fixed with 4% paraformaldehyde (PFA – 10 min) and permeabilized with 0.5% Triton X-100 (15 min) with thorough PBS washes in-between. Where cells were dried (see FISH methods), this only occurred following PFA fixation in order to preserve 3D structures and minimize cell and nuclear flattening.

## DNA FISH

A detailed method for DNA FISH has been described elsewhere (*Jubb and Boyle, 2020*). Briefly, DNA stocks of fosmid clones targeting EGFR (WI2-2910M03), CDK4 (WI2-0793J08), and PDGFRA (WI2-2022O22) (*Supplementary file 1*) were prepared via an alkaline lysis miniprep protocol (*Jubb and Boyle, 2020*). Each fosmid DNA probe was labelled via Nick Translation directly to a fluorescent dUTP (Green496-dUTP, ENZO Life Sciences; ChromaTide Alexa Fluor 594-5-dUTP, Thermo Fisher Scientific) and incubated with unlabelled dATP, dCTP, and dGTP, ice-cold DNase and DNA PolI for 90 min at 16°C. The reaction was quenched with EDTA and 20% SDS, TE buffer added, and the reaction mix run through a Quick Spin Sephadex G50 column.

Cells on slides or cover-slips were prepared by incubating for 1 hr in ×2 trisodium citrate and sodium chloride (SSC)/RNaseA 100 µg/ml at 37°C, then dehydrated in 70%, 90%, and 100% ethanol. Slides were warmed at 70°C prior to immersion in a denaturing solution (×2 SSC/70% formamide, pH 7.5) heated to 70°C (methanol:acetic acid-fixed cells) or 80°C (PFA-fixed cells), the duration of which was optimized to each cell line. After denaturing, slides were immersed in ice-cold 70% ethanol, then 90% and 100% ethanol at room temperature before air drying.

FISH probes were prepared by combining 100 ng of each directly labelled fosmid probe (per slide), 6 µg Human Cot-1 DNA (per probe), 5 µg sonicated salmon sperm (per slide), and 100% ethanol. Once completely dried, the resulting pellet was suspended in hybridization mix (50% deionized formamide [DF], ×2 SSC, 10% dextran sulfate, 1% Tween 20) for 1 hr at room temperature, denatured for 5 min at >70°C and annealed at 37°C for 15 min. Where relevant, FISH probes were instead hybridized in Chromosome 7 paint (XCP 7 Orange, Metasystems). The probes were incubated overnight at 37°C. The following day, the slides were washed in ×2 SSC (45°C), 0.1% SSC (60°C) and finally in ×4

SSC/0.1% Tween 20 with 50 ng/ml 4′,6-diamidino-2-phenylindole (DAPI). Slides were mounted with Vectashield.

## RNA FISH

RNA FISH probes (Custom Assay with Quasar 570 Dye) targeting the first intron (pool of 48 22-mer probes) of *EGFR* were designed and ordered via the Stellaris probe designer (Biosearch Technologies, Inc, Petaluma, CA) (https://www.biosearchtech.com/support/tools/design-software/stellaris-probe-designer, version 4.2). Cells were seeded, fixed, and permeabilized as above. Slides were immersed in ×2 SSC, 10% DF in DEPC-treated water for 2–5 min before applying the hybridization mix (Stellaris RNA FISH hyb buffer, 10% DF, 125 nm RNA FISH probe) for incubation at 37°C. After overnight incubation, slides were incubated in ×2 SSC, 10% formamide in DEPC-treated water for 30 min, and then stained with DAPI (5 ng/ml). Slides were washed with PBS before mounting with Vectashield.

## Combined RNA:DNA FISH

Nascent EGFR RNA FISH was performed as above, and nuclei imaged as described below. The x,y,z coordinates for each image were recorded via NIS software at the time of imaging. After removing the cover-slips and washing the slides in PBS, EGFR DNA FISH was performed whereby the probe preparation was as above. Centromere 7 (CEN7 – CHR07-Dig Control) FISH probe (Pisces Scientific) was prepared, denatured for 5 min at 80°C and snap-frozen on crushed ice. Slides were transferred from PBS wash to denaturing solution at 80°C for 15–30 min, washed in ×2 SSC, and incubated overnight with the probe(s) at 37°C. The subsequent stringency washes were as described above. Slides were then incubated in blocking buffer (×4 SSC/5% Marvel) for 5 min, followed by anti-digoxigenin antibody (Roche; 1 in 10; 1 hr at humidified 37°C) and anti-sheep Alexa Fluor 647 secondary antibody (Thermo Fisher Scientific; 1 in 10; 1 hr at humidified 37°C) with ×4 SSC/0.1% Tween 20 washes in between. After the final washes, slides were stained with DAPI and mounted as described above. The stored x,y,z coordinates were used to relocate and image each nucleus. Owing to the irregularity of the tumour nuclei, it was possible to be confident in re-imaging the correct nucleus – nuclei were excluded where this was not the case, or where nuclei were lost between RNA and DNA FISH. Spot counting was subsequently performed as described below with RNA and DNA foci being defined and counted separately to avoid influencing the outcome. For CEN7, nuclei were excluded if the number of foci could not be clearly identified.

## Immunofluorescence and immuno-FISH

Slides were blocked in 1%BSA/PBS/Triton X-100 0.1% for 30 min at 37°C before overnight incubation with the primary antibody at 4°C (Rpb1 NTD (D8L4Y) #14958, Cell Signaling Technology, 1 in 1000; mCherry [ab167453], abcam, 1 in 500). The following day, slides were washed in PBS before incubation with an appropriate secondary antibody (1 in 1000 Alexa Fluor) for 1 hr at 37°C. After further PBS washes and DAPI staining, slides were mounted with Vectashield.

For immuno-FISH (DNA), the IF signal was fixed via incubation with 4% PFA for 30 min. Following thorough PBS washes, the DNA FISH protocol was then followed as above.

For immuno-FISH (RNA), the antibodies were added at the same concentration as described above to the hybridization mix (primary antibody) and ×2 SSC/10% DF washes (secondary antibody).

## Flow cytometry and FACS

Cells were prepared by adding EGF-free media for 30 min before lifting and suspending cells in 0.1% BSA/PBS. Cells were incubated in 100 ng/ml EGF-647 (E35351, Thermo Fisher Scientific) in 0.1%BSA/PBS, with cells incubated in 0.1% BSA/PBS as a negative control, for 25 min. Cells were washed three times in 0.1%BSA/PBS before being analysed on the BD FACSAria III FUSION. Where indicated, cells were sorted by EGF-647 gated into high and low groups, and a sort check was performed to verify these were true populations prior to expanding these cells onto 22×22 mm$^2$ cover-slips. Fifteen days after the cells were sorted, the slides were fixed, permeabilized, and DNA FISH performed as above.

### mCherry_POLR2G knock-in cell line

crRNA and donor DNA was designed using the previously reported TAG-IN tool (*Dewari et al., 2018*), with the corresponding fluorescent reporter gene sequences for mCherry implemented into the existing tool (*Supplementary file 1*). Output sequences from the TAG-IN tool were manufactured by Twist Bioscience. Gene-specific crRNA (100 pmoles – IDT Technologies, Coralville, IA, USA) and universal tracrRNA (100 pmoles, IDT Technologies, Coralville, IA, USA) were assembled to a cr:tracrRNA complex by annealing at the following settings on a PCR block: 95°C for 5 min, step down cooling from 95°C to 85°C at 0.5°C/s, step down cooling from 85°C to 20°C at 0.1°C/s, store at 4°C. Recombinant Cas9 protein (10 µg, purified in house – see *Dewari et al., 2018*) was added to form the ribonucleoprotein (RNP) complex at room temperature for 10 min, then stored on ice; 300 ng of donor dsDNA were denatured in 30% DMSO by incubating at 95°C for 5 min followed by immediate immersion in ice. The donor dsDNA and RNPs were electroporated into E28 cells using the 4D Amaxa X Unit (programme DN-100). After 2 weeks of serial expansion of cells in 2D culture, assessment of knock-in efficiency was assessed by suspending $5–7 × 10^5$ cells in 0.2% BSA/PBS and analysed on BD LSRFortessa Cell Analyzer, with cells electroporated with tracrRNA:Cas9 only as a negative control. Cells were then further sorted into a pure KI population, and mCherry KI was verified by immunofluorescence for mCherry and Rpb1.

### Imaging

Slides were imaged on epifluorescence microscopes (Zeiss AxioImager 2 and Zeiss AxioImager.A1) and the SoRa spinning disk confocal microscope (Nikon CSU-W1 SoRa). For 3D image analysis, images were taken with the SoRa microscope and a 3 µm section across each nucleus was imaged in 0.1 µm steps. Images were denoised and deconvolved using NIS deconvolution software (blind preset or Lucy-Richardson) (Nikon). 3D images are shown in the figures as MIP prepared using ImageJ.

### Quantification and statistical analysis

#### Image analysis of nuclear localization

Images were analysed using Imarisv9.7 and Fiji. The scripts used to perform nuclear territory analysis have been described elsewhere (*Boyle et al., 2001*; *Croft et al., 1999*; see also Data availability). Briefly, single-slice images were taken with a ×20 lens using the Zeiss AxioImager 2, imaging at least 50 nuclei per cell line. The images were segmented first to individual nuclei, and subsequently the area of the DAPI signal was segmented to define the nuclear area. This area was segmented into concentric shells of equal area from the periphery to the centre of each nucleus. The signal intensity of each FISH probe or chromosome paint signal was calculated, with normalization for the DAPI signal in each shell.

#### Image analysis of ecDNA and large PolII foci

For 3D analysis, deconvolved images were analysed using Imaris (v9.7) and all analysis was performed on the full 3D image. RNA and DNA FISH foci, and where relevant, large PolII foci, were defined, counted and distances between them calculated, using the Spots function within Imaris. Imaris spot size diameter was selected by single plane measurement of representative foci and this defined diameter was applied to all nuclei of a given experiment for 3D analysis. For DNA FISH analysis, E26, E28, and E25 spot size was 300 nm diameter, and where indicated in the text, reanalysed with 150 nm spot diameter. For E20 and all RNA FISH experiments, a spot size diameter of 200 nm was used. For RPB1 and POLR2G foci (IF), large foci were defined as those ≥500 nm diameter (*Cho et al., 2018*; *Sabari et al., 2018*).

For 3D cluster analysis of FISH spots, Ripley's K function was performed using the x,y,z coordinates for each FISH spot using the Imaris Spots function to determine observed and null distribution values.

$$K(r) = \frac{1}{\lambda} \sum_{i \neq j} \frac{\mathbb{1}\left\{d(i,j) \leq r\right\}}{n}$$

Ripley's K function compares the number of points at a distance smaller than a given radius r, relative to the average number of points in the volume. This average is the density lambda, in this case the number of foci, n, divided by the volume. In the above equation,

is the indicator function which equals 1 if the distance between points i and j is no larger than r, and 0 otherwise. A high value of Ripley's K function represents clustering at the given radius r, whereas a low value represents dispersion. Consequently, a high Ripley's K function at a given radius is indicative of clustering at this radius. By comparing the observed value of Ripley's K function at a given radius with that computed on the same number of foci and with the same volume but drawn from a uniform null distribution, the presence of significant clustering in the given cluster at the given radius can be detected.

The code written to perform this analysis was formed using a script written in Python (v3.9) and has been made available on GitHub (see Data availability). Ripley's K function was determined across a radius of 0.1–1 µm in 0.1 µm increments. After calculating the observed Ripley's K function value, a null distribution of no clustering, estimated on uniformly distributed samples with the same number of spots, was generated using the coordinates for each given nucleus to calculate 10,000 Ripley's K function values at each radial increment. We tested a sample of nuclei with 50,000 values and confirmed that 10,000 values would provide sufficient accuracy. Having sampled that nucleus shape and size did not affect the significance of a result at each increment in the given range of radii, a bounding radius of 5 was used for all samples. Only nuclei with greater than 20 *EGFR* foci were included to ensure both that the majority of foci were ec*EGFR*, to allow adequate granularity and minimize the risk of a false negative result due to lack of foci. The p-value for each observed K function was established against the expected values using the Neyman-Pearson lemma. Where the observed and expected K function at p=0.05 were the same, a randomized binomial test was performed to determine if p<0.05 for the observed value, weighting the probability of success as the ratio of the number of values p<0.05 and the total number of equal values. Having determined this, the most optimistic estimate of p-value was made which would favour identification of a significant result, that is, a bias in favour of significant clustering. A Benjamini-Hochberg procedure was performed to control for the false discovery rate (FDR = 0.05).

All other statistical analysis was performed with GraphPad Prism v9.0. The statistical details for each experiment can be found in the relevant figure legends and in the Source Data. For figures, p-values are represented as follows: *<0.05, **<0.01, ***<0.001, ****<0.0001. Where appropriate, Bonferroni correction for multiple hypothesis testing was performed, and, where relevant, corrected p-values are those plotted in the figures and are given in the Source Data in brackets next to the uncorrected p value.

## RNA and WGS sequencing sample preparation, analysis, and processing

The preparation of these cell lines for RNA-seq has been described in detail elsewhere (*Gangoso et al., 2021*). WGS was undertaken by BGI Tech Solutions with PE100 and normal library construction. WGS, RNA-seq, and AmpliconArchitect data for GBM39 was taken from data made available via publication and in the NCBI Sequence Read Archive (BioProject: PRJNA506071) (*Wu et al., 2019*).

Sequences were aligned to hg38 with STAR 2.7.1a with settings '--outFilterMultimapNmax 1' used for WGS and RNA-seq data and settings '--alignMatesGapMax 2000 --alignIntronMax 1 --alignEndsType EndToEnd' used only for WGS data (*Dobin et al., 2013*). Duplicate reads were removed using Picard (Broad Institute). AmpliconArchitect (*Deshpande et al., 2019*) and Amplicon-Classifier (*Kim et al., 2020*) were used to predict the ecDNA regions and classify circular amplicons for E26 and E28, and to classify EGFR exons as being located primarily on ecDNA or only on chromosomal DNA in E26 and E28. Exon coordinates were extracted from Ensembl (isoform:EGFR-201, Ensembl Transcript ID: ENST00000275493.7). Alignments were converted to bigWig files using deepTools bamCoverage with setting '--normalizeUsingRPKM' (*Ramírez et al., 2016*) and visualized using the UCSC genome browser (*Kent et al., 2002*). HOMER2 (*Heinz et al., 2010*) makeTagDirectory and annotatePeaks.pl (settings '-len 0 -size given') were used for read counting of WGS and RNA in EGFR exons. Analysis of RNA-seq counts per copy number was performed using scripts written in Python (v3.9). We normalized the RNA-seq read counts to the WGS read count in each EGFR exon, and analysed in GraphPad Prism v9.0. SNP calling was done using strelka v2.9.10 (*Kim et al., 2018*) using the configureStrelkaGermlineWorkflow.py command on all samples (WGS blood, WGS tumour, and RNA-seq tumour) for each cell line (E26 and E28) separately. SNPs that passed all filters were extracted using bcftools (*Danecek et al., 2021*) and selected for those that had an allele frequency in the WGS

blood between 40% and 60%. The ratio of allele frequencies between the RNA-seq and WGS tumour samples were determined for those SNPs overlapping expressed exons with at least 20 reads in the RNA-seq samples . See Data availability.

## Source data

Source data regarding the statistical tests applied, the exact sample number, p-values of tests (and adjustments for multiple hypothesis testing), and details of replicates are included where indicated in the article. N=number of nuclei.

## Acknowledgements

SVB would like to thank Dr Tim Cannings for helpful suggestions on statistical analysis.

We acknowledge the Advanced Imaging Resource at the Institute of Genetics and Cancer and the Edinburgh Super-Resolution Imaging Consortium (ESRIC), and the Flow Cytometry team at the Centre for Regenerative Medicine, University of Edinburgh, for their technical support. This work has made use of the resources provided by the Edinburgh Compute and Data Facility (ECDF) (http://www.ecdf.ed.ac.uk/). KP was supported by a Wellcome PhD Training Fellowship (220399/Z/20/Z). ETF was supported by the Swiss National Science Foundation (P2ELP3_191695). GMM and the Glioma Cellular Genetics Resource (https://www.gcgr.org.uk/) were supported by the Cancer Research UK (CRUK) Centre Accelerator Award (A21922). AH was supported by a CRUK PhD Fellowship (C157/A29279). SMP is a Cancer Research UK Senior Research Fellow (A17368). Work in the group of WAB is supported by MRC University Unit grant MC_UU_00007/2.

Funding sources were not involved in study design, data collection, data interpretation, or the decision to submit the work for publication.

## Additional information

### Funding

| Funder | Grant reference number | Author |
|---|---|---|
| Wellcome Trust | 220399/Z/20/Z | Karin Purshouse |
| Swiss National Science Foundation | P2ELP3_191695 | Elias T Friman |
| Cancer Research UK | DRCNPG-Nov21\100002 | Steven M Pollard |
| Cancer Research UK | C157/A29279 | Alhafidz Hamdan |
| Cancer Research UK | A17368 | Karin Purshouse |
| Medical Research Foundation | MC_UU_00007/2 | Wendy A Bickmore |

The funders had no role in study design, data collection and interpretation, or the decision to submit the work for publication. For the purpose of Open Access, the authors have applied a CC BY public copyright license to any Author Accepted Manuscript version arising from this submission.

### Author contributions

Karin Purshouse, Investigation, Methodology, Writing – original draft, Writing – review and editing; Elias T Friman, Sjoerd V Beentjes, Formal analysis, Methodology, Writing – review and editing; Shelagh Boyle, Pooran Singh Dewari, Vivien Grant, Gillian M Morrison, Investigation; Alhafidz Hamdan, Formal analysis, Investigation; Paul M Brennan, Resources; Steven M Pollard, Wendy A Bickmore, Conceptualization, Supervision, Funding acquisition, Writing – original draft, Project administration, Writing – review and editing

### Author ORCIDs

Karin Purshouse (iD) http://orcid.org/0000-0003-0942-6342

Alhafidz Hamdan http://orcid.org/0000-0002-0794-5504
Sjoerd V Beentjes http://orcid.org/0000-0002-7998-4262
Steven M Pollard http://orcid.org/0000-0001-6428-0492
Wendy A Bickmore http://orcid.org/0000-0001-6660-7735

**Decision letter and Author response**
Decision letter https://doi.org/10.7554/eLife.80207.sa1
Author response https://doi.org/10.7554/eLife.80207.sa2

## Additional files

**Supplementary files**
• Supplementary file 1. Genomic information for FISH probes and CRISPR knockin. (A)Fosmid probes for DNA FISH related to STAR methods. Genome coordinates (Mb) are from the hg38 assembly of the human genome. (B) CrRNA sequence and dsDNA sequence for mCherry_PolR2G CRISPR knock-in.

• MDAR checklist

**Data availability**
WGS and RNAseq data have been deposited on NCBI GEO under study accession number GSE215420 and is publicly available as of the date of publication As indicated in the Key Resources, all original code has been deposited as: https://github.com/IGC-Advanced-Imaging-Resource/Purshouse2022_paper (copy archived at swh:1:rev:5b1a3920afa8e85132c94bcc6dfce94575f939ce) https://github.com/SjoerdVBeentjes/ripleyk (copy archived at swh:1:rev:1303af539403303786b-6460fabef355ea345ea6c9) https://github.com/kpurshouse/ecDNAcluster (copy archived at swh:1:rev:9162a39f3c8e19e973eaedc50ad4e1d3dc570e90).

The following dataset was generated:

| Author(s) | Year | Dataset title | Dataset URL | Database and Identifier |
|---|---|---|---|---|
| Purshouse et al | 2022 | WGS and RNA-seq data E26,E28 | https://www.ncbi.nlm.nih.gov/geo/query/acc.cgi?acc=GSE215420 | NCBI Gene Expression Omnibus, GSE215420 |

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
