## [Editor Report]

This study convincingly shows that, in contrast to recent reports, the transcriptional output of oncogenes carried on extrachromosomal DNA (ecDNA) in glioblastoma cell lines is driven by the copy number of the ecDNA, rather than their spatial localization into transcriptional hubs. This study is relevant to researchers interested in nuclear function, particularly transcriptional organization within malignant cells.

---

## [Decision Letter]

[Editors' note: this paper was reviewed by Review Commons.]

---

## [Author Response]

General Statements

We thank the reviewers for the detailed and considered comments. We believe they have significantly improved and enhanced the manuscript.

Point-by-point description of the revisionsReviewer #1 (Evidence, reproducibility and clarity (Required)):The manuscript by Purshouse et al. is focused on the question whether oncogene expression from amplified extrachromosomal DNA (ecDNA) is driven by their intranuclear positioning and special transcriptional control. Using microscopy after FISH or immuno-FISH with following quantitative analysis, as well as RNA and WGS sequencing, the authors show that transcription output of oncogenes in the three studied glioblastoma cell lines depends primarily on the ecDNA copy number but not on spatial localization and formation so called transcriptional hubs, as has been claimed in recent publications.Overall, the study is very well conceived and presented, experimental approaches are adequate and carefully controlled. The presentation of the results is clear, the images are of a high quality, the data are comprehensively discussed and the paper is agreeable to read.I have only several small comments to the authors:(1) Figure 1D,F and Figure S1C:Classification of signals into peripheral and internal is extremely difficult in adherently growing flat cells, such as glioblastoma cells. This fact is aggravated by performing a 2D analysis of signal distribution within concentric shells. Therefore I wonder how the authors can exclude that signals in the central shells are not sitting at the top or bottom of the nuclei, i.e., are not peripheral?

Analysis of 2D images to infer 3D organisation is a well-established approach. Though indeed, in a single nucleus a signal in the central shells might be on the top or bottom surface, averaging over many nuclei (assuming random orientation with respect to the slide surface) allows identification of a truly central preferential localisation (references included in the methods).

Boyle, S., Gilchrist, S., Bridger, J.M., Mahy, N.L., Ellis, J.A., and Bickmore, W.A. (2001). The spatial organization of human chromosomes within the nuclei of normal and emerin-mutant cells. Hum. Mol. Genet. 10, 211–219.Croft, J.A., Bridger, J.M., Boyle, S., Perry, P., Teague, P., and Bickmore, W.A. (1999). Differences in the localization and morphology of chromosomes in the human nucleus. J. Cell Biol. 145, 1119–1131

Please note: Images were taken as a single plane image through a nucleus, and not as a stacked 3D image. Although GBM cells are relatively flat when grown as adherent cells, they still represent a 3D structure and foci in the top or bottom of the nuclei would not have been in focus in a 2D single plane image. In addition, our finding of ecDNA occupying a more central territory is consistent with previous publications, which we have cited.

(2) Although the authors note a significant reduction in peripheral localization of EGFR signal in nuclei of glioblastoma cells in comparison to control NSCs, the signal remain to a great degree peripheral – e.g. Figure S1C, Figure 2D, Figure 3D, Figure 4D (also see the previous comment). I am wondering whether it could be explained by sticking of ecDNA to periphery of early/late anaphase chromatin during mitotic segregation? Have the authors observed ecDNA positioning at these cell cycle stages?

We considered exploring ecDNA positioning at different cell cycle stages, particularly whether quiescent GBM cells would have altered ecDNA regulation. However, we felt this represented a separate important research question that is beyond the scope of the current study. Moreover, cell cycle control would not alter our main conclusions. The patient-derived GBM models we have used here are slow growing and so very few cells will be in anaphase. In a culture of GSCs we anticipate approximately only ~1% are in mitosis – so this would have a limited contribution to the current data.

(3) The Figure 1A shows an example of HSR, formation of which is the next step in evolution of ecDNA elements in tumor cells. Have the authors observed HSR in interphase nuclei? And if yes, what is their location – peripheral or internal?

There is no means of confirming that a group of EGFR DNA FISH foci represent an HSR rather than a group of clustered ecDNA (or double minutes) in an interphase nucleus. Consequently, as our study sought to evaluate the presence or absence of clustering of ecDNA, we were mindful of the possibility of HSRs in the E26 cells resulting in a ‘false positive’ clustering effect in our later cluster analysis. We decided for this reason to include cell lines, E28 and E25, where no HSRs were observed in metaphase spreads. This is a good point, however, and we therefore make this more explicit in the current results.

(4) First paragraph of the chapter "EcDNA do not cluster in the nucleus":"We used 3D image-based analysis of the EGFR DNA FISH signals…"What was the thickness of studied nuclei after hybridization? As far as I understood from the Methods, nuclei were air-dried during FISH procedure, a step which significantly flattens cells including nuclei. My concern is whether the analysis the authors performed is a real 3D analysis. I do not see a problem if it is 2D, but the limitations of the method have to be mentioned in any case.

Cells were only air dried after PFA fixation so we believe the 3D structure will therefore be largely preserved (see Methods). As noted in the methods section, nuclear sections were imaged in 3D across 3uM in 100nm increments in the z plane. Even if flattening during processing did occur, this would have increased the likelihood of observing close clustering across the cross-section of each nucleus imaged.

We have added a note to the Methods ‘Metaphase spreads and interphase nuclei’ to make this methodology clear, and direct the reviewer to the Methods section ‘Imaging’ where the 3D imaging strategy is outlined.

(5) The same chapter: "…if there were clustering of ecDNA in the close proximity required for coordinated transcription in condensates or hubs; this should be ~200nm or less." I think it is important to explain why the authors have chosen this threshold of 200 nm.

We have now added a sentence to this paragraph (Results, ‘EcDNA in GBM stem cells do not cluster in the nucleus’) to provide a referenced justification for this threshold.

We chose this threshold based on optical resolution for conventional wide-field microscopy being ~200nm. The resolution of the SoRa super-resolution microscope used in our study is approximately 120nm. We have added this to the Discussion so this is clearer for the reader.

200nm has been used as a threshold in published studies (see cited) for colocalisation when exploring genomic loci proximity and contact domains, and previous work indicating interactions between super-enhancers and BRD4/Med1 puncta appeared to occur within this distance. So we believe this is appropriate to enable cross-comparison of existing literature.

In similar spirit to this point, we felt greater explanation around choice of PolII hub size and how foci were measured using the Spots function in Imaris would be beneficial to the reader. We have added this to the ‘Image analysis of ecDNA and large PolII foci’ section of the methods.

(6) The same chapter: the authors consider a single FISH signal (a spot) as a single ecDNA element. Why the authors are sure that these spots are not small clusters themselves? From the figures through the paper, I can see that FISH spots do very in size.

The use of a heterotypic ecDNA-harbouring cell line (E25) is central to address this concern. We have analysed another heterotypic ecDNA-harbouring cell line (E20) also in response to Reviewer 2; Point 1b.

Due to the variable nature of ecDNA structure, we agree that some FISH foci are smaller due to different ecDNA sizes and possibly different break points in individual ecDNA. To address these points, we have added the following to the manuscript (Discussion, para 3).

We have performed reanalysis of the raw data. The largest ecDNA EGFR FISH foci were similar in size to those on the native chromosomal signal in NSC cells, which suggest these are not multiple clusters of smaller foci. It should be noted that the images in figure 1 were taken with an epifluorescence microscope with a lower optical resolution appropriate to the erosion territory analysis, and this may overstate the presence of large ecDNA loci. To ensure we were minimising the risk of missing two closely localised elements, we performed all subsequent analysis in 3D with images taken on the SoRa microscope (optic resolution ~120nm).

We use two-colour FISH to visualise two ecDNA populations in two cell lines. This is an important control for the possibility of small clustering, as spots from the two different populations would be expected to cluster in this scenario. In the E25 cell line we repeated the analysis by using a 150nm size threshold in Imaris (i.e. half the size) to ensure there was no omission of such small foci in small clusters, and observed the same result (Figure 3 Supplemental Figure 1).

We repeated the analysis in greater numbers in another dual-ecDNA population cell line, E20, where metaphase spread showed that a small proportion of ecDNA (~10%) had both oncogenes on the same ecDNA (Figure 3 Supplemental Figure 2). A similar proportion were identified as colocalising via Ripley’s K. This gave us confidence both in our ability to capture true colocalisation, and that the lack of close clustering in all other interphase nuclei in this cell line was true.

(7) Figure S5D: on the RGB images DNA signals are hardly or not visible at all. I suggest the authors to convert the signal channel into enhanced grey scale image and, instead of showing counterstain with DAPI, outline nuclear borders.Similar comment is for Figure S5G and Figure 5B: either use grayscale images for the signals or enhance the RGB signal channels so that it is clearly visible on the DAPI background.

Good suggestion. Images for these figures are now presented in greyscale for easier visualisation. For figure 5A we have supplied the images in full colour (as these now include new data (CEN7 FISH) – we felt black and white images in this instance were not as clear). We are happy to supply this version if preferred on review.

(8) There are several textual inaccuracies:"…was confirmed by DNA FISH on metaphase chromosomes…" – the authors probably meant "metaphase spreads""…10% formamide in DEPC-treated water for 30 min, and then repeated with DAPI…" – the authors probably meant "stained with DAPI"For clarity, the ordinate axis in Figure 1F has to be relabeled as "Normalised EGFP:chr 7 mean signal intensity"

Agree. The text and figure legend have been amended.

Reviewer #1 (Significance (Required)):This paper is a good example of a report about negative results contradicting recently published data that became a common wisdom. The present work is important for our understanding of nuclear functioning and, in particular, of transcription organization within malignant cell nuclei. I fully support publication of this work.Reviewer #2 (Evidence, reproducibility and clarity (Required)):In this study, Purshouse et al. performed super-resolution microscopy to (1) investigate the cytogenetic features of amplified oncogenes; and (2) assess the quantitative relationship between oncogene expression and DNA copy number. The authors found that extra chromosomally amplified DNA in cancer cells does not form transcriptional condensates and the transcriptional output of amplified DNA is largely proportional to the DNA copy number. These findings suggest that high-level gene amplification is sufficient to drive transcriptional amplification.I. Regarding the 1st conclusion that ecDNAs (also referred to as double-minute chromosomes or DMs) do not form condensates, the authors provide two lines of evidence. First, individual ecDNA circles containing the same amplified DNA do not cluster together (Figure 2). Second, ecDNA circles containing different amplified DNA also do not cluster together (Figure 3). Both observations were quantified by the average distance between ecDNA circles detected by DNA-FISH. If the authors had only shown that ecDNA circles containing the same DNA do not cluster together, this could not rule out the possibility that some FISH spots represent tightly clustered DMs/ecDNAs within 200nm that cannot be resolved by microscopy. However, the observation that different ecDNA circles (which can be resolved using different FISH probes) do not co-localize with each other provides compelling evidence for the lack of apparent ecDNA/DM clustering. Overall I consider these data convincingly support the conclusion that ecDNAs/DMs do not form condensates. I have one question about Figure 2 and one suggestion.Ia. How did the authors distinguish between large FISH spots and two clustered FISH spots?

Please see also response to Reviewer 1, Question 6. We feel the two-colour FISH was essential in controlling for this possibility, and refer you further to our response to your comments to 1b below.

Can the authors distinguish between large FISH spots representing ecDNA condensates and those reflecting HSRs (homogeneously staining regions)?

Please also refer to answer to Reviewer 1, Point 3. Due to this concern, we made sure to include cell lines that harboured entirely (E28, E25) or mostly (E26) ecDNA, and the additional cell line, E20, also appeared to harbour only ecDNA. Had we observed close clustering, our first question would have been whether these were truly clustered ecDNA or HSRs. In addition, a criterion of our Ripley’s K function was that we only included nuclei with >20 loci (See methods), which would have likely excluded any nuclei in which only an HSR was present.

Ib. To further strengthen the conclusion, the authors may perform the same analysis on at least another cell line with multiple ecDNA species. (Currently this was only done on the E25 cell line.) In my opinion, this experiment is not essential but will significantly strengthen the conclusion.

We thank the reviewer for this suggestion, and have performed the same analysis on another cell line, E20, that harbours CDK4 and PDGFRA ecDNA. Interestingly, we observed ~10% ecDNA had colocalised CDK4 and PDGFRA on metaphase spreads, suggesting the two genes are on the same ecDNA molecule. In interphase nuclei we observed a similar proportion of interphase nuclei with this pattern, and Ripley’s K confirmed these nuclei had colocalised CDK4/PDGFRA foci. This gives confidence that 3D Ripley’s K is able to identify true colocalisation. We note that no other nuclei (22/24) had significant clustering at <300nm when considering CDK4 and PDGFRA together.

We note in addition that 4/24 nuclei had clustering of CDK4 foci at 200nm. We reviewed these nuclei, and believe that these represent double minutes, and have included a representative image in the figures.

We have added these data to Figure 3 and Supplementary Figure 3, and to the text in the Results and Discussion.

The authors performed additional experiments showing that ecDNAs/DMs do not co-localize with transcriptional condensates (Figure 4). Given the somewhat ambiguous criteria for the identification of transcriptional condensates and the promiscuity of their biological nature (see for example, the review of McSwiggen et al., Genes & Dev. 2019), I consider these data to be informative though not as convincing as the direct analysis of ecDNA/DM clustering.

We acknowledge the reviewer’s point and have amended the Results subheading from ‘condensates’ to ‘hubs’ i.e. *Transcriptional hubs are few and do not colocalize with ecDNA in GBM stem cells’.* We have altered ‘condensates’ to ‘large PolII foci/hubs’ as appropriate throughout the manuscript to be more specific.

We have clarified our description of the literature around ecDNA and colocalisation with RNAPolII (Discussion, para 5). It is inherently challenging to control for varying ecDNA number and chance colocalisation of ecDNA and PolII foci, hence our focus on large PolII foci. We have endeavoured to clearly outline our methods (see last bullet point of response to reviewer 1, point 5) so this is clear to the reader and have added text to highlight the limitations in the discussion.

II. Regarding the 2nd conclusion that the transcriptional output of amplified DNA is largely proportional to the DNA copy number, the authors assessed both the transcriptional efficiency (nascent RNA FISH) and the transcriptional yield of amplified ecDNA (RNA-Seq). To assess transcriptional efficiency (Figure 5), the authors performed dual DNA/RNA-FISH to (1) measure the frequency of co-localization of amplified oncogene (EGFR) and nascent RNA; (2) compare the frequency of active transcription of extrachromosomal and intrachromosomal copies of the same oncogene. I have a few comments about these data and their interpretation.IIa. Figure 5B and 5C suggest that the transcriptional efficiency of ecDNA circles is dependent on the sequence of amplified DNA (as it varies between different glioblastoma lines) but not on the number of ecDNA circles (demonstrated by the linear relationship between actively transcribing and total ecDNA). This observation provides a great example showing that the genetic sequence of amplified DNA plays a big role in the transcriptional output. This example demonstrates that one cannot naively attribute differential gene expression in different cancer cell lines (even after normalization of gene copy number) to putative epigenetic changes and ignore the genetic variation.IIb. The linear relationship between DNA and RNA foci in Figure 5C does not exclude the possibility that tightly clustered ecDNAs/DMs, which only produce single DNA foci, may be transcribed more frequently. The authors should comment on this.

There is the possibility that we are unable to resolve multiple clustered DNA and RNA foci. We refer to our comment to Reviewer 1, Question 6, and note our addition to the text highlighting this caveat (Discussion, para 3). We have increased the number of biological replicates (n=3 – see new Figure 5B) and performed further RNAseq/WGS analysis (see response to IId. below) of SNPs which we feel significantly adds to this section of the study.

IIc. In Figure 5D, the authors analyzed the transcriptional efficiency of intra-chromosomal and extra-chromosomal EGFR gene copies. The heuristic classification of intra- versus extra-chromosomal EGFR gene copies (based on the number of total EGFR foci in a single cell) is less than satisfactory. I wonder whether the author could redo the experiment with additional FISH probes against either Chr7-centromeric DNA or another gene on Chr7p next to the endogeneous EGFR locus to determine the endogeneous EGFR loci with better certainty. This will enable a direct measurement of the transcriptional efficiency of endogenous EGFR and extrachromosomal EGFR in the same cell.

We agree, and have repeated this experiment (with 3 biological replicates) with a Centromere 7 control probe (See figure 5A for representative images), and included these data in evaluating overall RNA:DNA FISH ratios (Figure 5B). We then used these data to correlate RNA:DNA FISH ratio against the proportion of ecDNA (RNA:DNA ratio = number of RNA foci / number of DNA foci. EcDNA proportion = (number of *EGFR* DNA foci – number of CEN7 foci) / number of *EGFR* DNA foci) by Spearman’s correlation. The data from these technical replicates is included in Figure 5 – Supplementary file 1, and data from replicate 1 is shown in figure 5C. We agree this is a more accurate measure of ecDNA transcriptional efficiency, and hope this is clearer for the reader. We have removed the previous figures and supplemental figure, and amended the methods, results and discussion to reflect this.

IId. In Figure 5E-G, the authors further analyzed the transcriptional yield of endogeneous EGFR and amplified EGFR derived from RNA-Seq data. The authors used a clever trick to separate the transcriptional output of wild-type EGFR from amplified EGFR vIII (with exon 2-7 deletion). The result supports the conclusion that the normalized transcriptional yield of the deleted exons is similar to that of amplified exons. But the large variation in the RNA:DNA ratio of non-deleted exons in the ecDNA group is puzzling. I suspect this may be due to the size variation of individual exons.The authors described the analysis as follows in the Method section:WGS and RNA-seq read counts were normalised to the size of the ecDNA/chromosome blocks and EGFR exons, respectively. Normalized RNA-seq read counts of each exon were divided by the normalised WGS read counts of the corresponding ecDNA/chromosome region to give a normalized RNA-seq count for each exon, and analysed in Graphpad Prism v9.0.Why is it necessary to normalize WGS/RNA-Seq counts to the size of ecDNA blocks or exonic size? (1) One can simply normalize the RNA read counts to the DNA read counts in each exon and compare the ratio across different exons. The ratio automatically controls for the exonic size, sequencing depth, etc. (2) A potentially more specific analysis is to calculate the ratio of RNA reads joining exon 1 to exon 8 (from EGFR vIII) and RNA reads joining exon 1 to exon 2 (from wildtype EGFR), and compare the RNA ratio to DNA ratio calculated from the average read depth in EGFR minus the average read depth in exons 2-7 (EGFR vIII) and the average depth in exons 2-7 (wildtype EGFR).

We agree that counting reads in individual exons is a better way to perform the intended analysis. We have performed this analysis in the E26 and GBM39 cell lines directly comparing WGS and RNAseq counts in exons 1-28 (with exons 2-7 = chromosomal, exons 1, 8-28 = predominantly ecDNA). As expected, the result is comparable to that of our previous method. We have replaced the AmpliconArchitect block analysis with this exon-based analysis in the manuscript and methods.

To confirm these new data were comparable to our previous analysis, we performed correlation analysis on the normalized RNA counts for the original Amplicon Architect (AA) vs revised exons approach. All correlations were positive (Pearson correlation, p<0.05 in all cases – plots shown in Author response image 1).

**Author response image 1. sa2fig1:** 

Regarding using exon 1-2 and 1-8 spanning reads as a specific measurement of extrachromosomal and chromosomal DNA, we were interested to perform this analysis. However, we noted technical issues that limit this. Notably, exon 1 of EGFR sits at the TSS which contains a CpG island. We noticed that there is a drop in sequencing read coverage here and genome-wide across CpG islands in the WGS data due to GC-bias in the sequencing or sample preparation. Furthermore, the polyA-enriched RNA sequencing data is heavily biased toward the 3’ end of the gene. See Author response image 2. These two technical biases mean that the comparison of ratios of spliced reads to genome coverage at exons will not be meaningful. We note that these biases could also influence the RNA/DNA ratio at individual exons (Author response image 1), which likely explains some of the variability in normalised RNA-seq counts between exons. However, the chromosomal exons 2-7 are toward the 5’ end of the gene, so if anything their RNA-seq reads would be underestimated compared to the mostly extrachromosomal exons >7. Importantly, the allelic ratio of SNPs would not suffer from these biases, as ratios of RNA and WGS reads across individual SNPs are compared directly to each other and single basepair changes are unlikely to have a technical effect on sequencing coverage. See Author response image 2 for this analysis.

I suggest two additional calculations that can further strengthen this analysis. (1) If there are polymorphic sites in the amplified sequence, the authors can calculate the fraction of allele-specific transcripts and the fraction of allele-specific genomic DNA from the sequencing data, and then calculate the RNA:DNA ratio. As the amplified DNA is derived from one parental homolog, this analysis can be done even in GBM lines with amplified wtEGFR. (2) If there are other co-amplified genes, the authors can perform allele-specific RNA:DNA analysis on those genes. This analysis will be informative as the co-amplified gene(s) may not be under positive selection.

We thank the reviewer for this suggestion which we believe strengthens our conclusions. We called germline variants in patient control (blood) samples using strelka2 and selected expressed exon-overlapping polymorphisms in the amplified region to calculate the allele frequencies in the WGS and RNA-seq samples (added as Figure 5 Supplementary Figure 5F). In the E26 cell line, only EGFR and LANCL2 (200kb 3’ of EGFR) are sufficiently expressed and have overlapping polymorphisms present on ecDNA (note only a proportion of LANCL2 polymorphisms are present on a subset of ecDNA). Nevertheless, the allelic ratio in the DNA/RNA is close to one for all polymorphisms. We also performed the same analysis on E28, which harbours expressed SNPs in both LANCL2 and VSTM2A, also showing allelic ratios close to 1, in line with our previous analysis. These data have now been added to the results and Figure 5D.

The lack of non-tumour patient controls prohibits us to call such polymorphisms in the GBM39 cell line.

Additional comments:IIe. There is apparent variation in the intensities of both DNA-FISH and RNA-FISH (see e.g., Figure 5A). Does this reflect copy-number variation of amplified DNA in single ecDNA/DM? Can the authors quantify such variation as well as its correlation with the efficiency of transcription?

We have subjectively not noted a correlation between RNA FISH signal intensity and its associated DNA FISH locus. We have now undertaken a pilot comparison of RNA/DNA FISH foci signal intensity in E26 and E28 cell lines using the raw data from Figure 5A. We observed no correlation in this small sample. Based on this pilot analysis, we do not believe further assessment of this would be useful or alter our main conclusions. We have also commented on the variation in the text (See Reviewer 1 Point 6).

There are always slight variations in intensity due to differences in how a FISH probe accesses the nucleus. In addition, differences in these images may reflect how these representative images were prepared for the manuscript (i.e. differences between cell lines). Raw imaging data can be made available and may further reassure the reviewer.

III. I suggest the authors cite more original research papers related to ecDNA/double-minute amplification.Original report of double-minute chromosomesCox, Yuncken, Spriggs, Lancet, 1965.EM analysis of double-minutesHamkalo et al., PNAS 1985.First cytogenetic analysis of extrachromosomally amplified EGFR in primary glioma:Vogt et al., PNAS 2004First study showing complex double minutes composed of multiple non-syntenic segments in a single tumorGibaud et al., Hum Mol Genet 2010.Dynamic evolution of multiple oncogenic ecDNA/DMs in glioblastomas:Snuderl et al., Cancer Cell 2011.Szerlip et al., PNAS 2012.First single-cell genomic analysis of amplified DNA in glioblastomas:Francis et al., Cancer Discovery 2014

We appreciate these recommendations. We have incorporated the following references where appropriate into the introduction (as they appear chronologically in the text):

Snuderl et al., Cancer Cell 2011.

Szerlip et al., PNAS 2012.

Cox et al., Lancet, 1965

Hamkalo et al., PNAS, 1985

Vogt et al., PNAS, 2004

We have added a sentence associated with these references: ‘EcDNA can be composed of multiple genetic fragments generated as a result of chromothripsis’

Gibaud et al., Hum Mol Genet, 2010

In association with this we felt it was important to add two further key references to ecDNA formation/structure:

Shoshani et al., Nature, 2021

Rosswog et al., Nature Genetics, 2021

We have not included the Francis et al. reference. While this is an important original research paper, this focuses more on EGFRvII, which is not of central relevance to this paper. Cross-commenting:

I agree with most comments from the other two reviewers and am happy to discuss my comments if needed.Reviewer #2 (Significance (Required)):Gene amplification and double-minute chromosomes were both discovered more than 50 years ago. But the molecular and genetic features of amplified DNA, including their origin, remain incompletely understood. There is a resurgence of interest in extrachromosomally amplified DNA (originally termed double-minute chromosomes) in cancer cells due to their connection to tumor evolution and therapy resistance. Several recent studies showed that ecDNA/DMs have unique chromatin organization and epigenetic states that promote gene transcription. Some of these studies suggested that epigenetic alterations, including enhancer rewiring, may be more important than putative gene copy-number amplification. Contrary to these studies, the current study suggests a more classical (genetic) and more intuitive model of amplified DNA, highlighting the determining contribution of gene copy number to transcriptional output. In comparison to other ecDNA papers, the current study does not have as many "novelty" factors, such as the usage of novel sequencing techniques or the proposed novel mechanisms of transcriptional upregulation (e.g., transcriptional condensates/hubs). However, I have found the data and analyses presented in the current study to be more convincing. This is because of two reasons. First, amplified DNA is genetically unstable and often displays cell-to-cell variation. Such heterogeneity cannot be resolved by bulk measurements, including DNA, RNA, epigenome, or Hi-C sequencing. By contrast, single-cell imaging can directly capture such variation. Second, the unique epigenetic/transcriptional features of ecDNA were often inferred from data generated using complementary assays (including different bulk sequencing measurements and imaging) on different cells. These experimental assays are often subject to different technical or biological variations that are difficult to control. By contrast, the current study performs simultaneous measurements of the number and transcriptional efficiency of ecDNA/DMs using a single experimental assay (super-resolution FISH), which is a more robust strategy.As my expertise is in genomic analysis, I know very well the limitations of sequencing assays and bioinformatic analysis, especially in the resolution of genetic/epigenetic features of repetitive DNA. For example, one cannot distinguish between extra-chromosomal (ecDNA/DM) or intra-chromosomal (homogeneous staining regions) amplified DNA based on bulk sequencing data, despite claims made in some recent studies (such as Wu et al. 2019). This is both because of the dynamic conversion between DMs and HSRs and because rearrangement junctions detected from sequencing data may be either within a single copy of amplified DNA (as in DMs) or between different amplified DNA copies as in HSRs. As my research focuses on chromosomal instability and cancer genomic rearrangements, I also have good knowledge of cancer genomics, cytogenetics, and mechanisms of chromosomal instability. Overall I have found the authors' analysis to be solid, although the conclusion could be strengthened with the analysis of more samples and with a comparison between ecDNA/DMs and HSRs.

We thank the reviewer for their detailed comments. We agree with the reviewer’s comment regarding more samples and have outlined above the specific additional experiments above which we agree significantly strengthens the analysis.

Regarding ecDNA vs HSRs, we refer the reviewer to our comments to Reviewer 1, Question 3. The data presented here cannot, and do not seek to, make a comparison between HSRs and ecDNA, and indeed have sought to ensure all analysis focuses on ecDNA characteristics. We make this clear in the updated manuscript. Exploration of HSRs and ecDNA transitions, as well as cell cycle *impact, are areas ripe for future studies.*

Reviewer #3 (Evidence, reproducibility and clarity (Required)):Recent work highlights the importance of ecDNA in cancer, including its role in enhanced oncogene transcription. Two independent studies recently demonstrated that ecDNA can form hubs or clusters, and in doing so, leads to further enhanced transcription. The authors are addressing an important problem. Understanding when hubs do, and do not form, and examining the context cell lineage, copy number, and other contexts, including cell cycle, is an important area of study. Further, these entities appear to be dynamic, as opposed to static. Therefore, there is considerable merit in studying this topic.The authors have characterized a set of glioblastoma neurosphere models, using spatial, transcriptomic, and computational methods to conclude that ecDNA may not require hubs to drive oncogenic transcription, which can result purely from elevated copy number driven by non-chromosomal inheritance. There is no doubt that the elevated copy number plays a key role, however, understanding this complex behavior related to hubs is worthy of study.A number of major concerns are raised:1) The title doesn't seem to cover the scope of work. It really is an analysis of GBM stem cell cultures, not a broad analysis of cancer. The title should be more reflective of the work.

We provide a revised title: ‘Oncogene expression from extrachromosomal DNA is driven by copy number amplification and does not require spatial clustering in glioblastoma stem cells’

2) The models used are described as being characterized by the GCRC, but a trip to the website to try to learn more about the models indicates: "COMING SOON". This needs to be rectified.

We agree, and the GEO Accession numbers will be added to the Key Resources so all key data relating to this manuscript are available at the point of publication. We note that the GCGR are characterizing these cell lines as part of an upcoming resource, so have also referenced this in the Key Resources.

3) If the authors do not believe that hubs play any role, then the right experiment is to analyze cancer cell line models that have been studied by others and shown to have hubs. If the authors do not want to do that, they need to modify their claims and restrict them to the models they studied.

We propose the following to make our claims in clear connection to the models studied (in addition to the title change noted above), and have amended the Results sections headers as follows:

EcDNA do not cluster in the nucleus -> EcDNA in GBM stem cells do not cluster in the nucleusTranscriptional condensates are few and do not colocalize with ecDNA -> Transcriptional condensates are few and do not colocalize with ecDNA in GBM stem cellsLevels of transcription from ecDNA reflect copy number but not enhanced transcriptional efficiency -> Levels of EGFR transcription from ecDNA reflect copy number but not enhanced transcriptional efficiency

We have checked the manuscript for other opportunities to make clear the models used e.g. ‘*in GBM stem cells’*, see Discussion para 1 and para 5. We have added a point of clarification the Results paragraph ‘Transcriptional hubs….’ – ‘We next assessed whether ecDNA foci, which, whilst not clustered, share regional localization, …..’ (see response to Reviewer 3, Question 5 below).

We carefully considered our approach to address the question of ecDNA transcriptional hubs, namely undertaking a 3D approach with mathematical modelling to control for ecDNA number and nuclear size. We did not feel that the methods used in previously cited papers (2D correlation analysis) were suitable for evaluating 3D clustering at transcriptionally relevant distances in nanometre distances rather than pixels. We have clearly outlined the aims and models used, made the code openly available and made sure our claims are clearly stated for the reader to consider accordingly. Indeed the addition of another cell line (E20) in which some ecDNA have CDK4 and PDGFRA colocalised (in both metaphase spread and interphase nuclei in similar proportions) which was captured via 3D Ripley’s K gives us confidence in our tool. We hope this will be a useful resource and set of analytic tools for the field.

We have outlined through the discussion why we our results may differ, and why our approach cannot necessarily be used in other cell lines. For example, COLO320 harbours 3 copies of MYC on each ecDNA, and the ecDNA are very large (Discussion, para 4).

4) In previous studies, ecDNA hub formation was found to occur in about half of the cells over a 48 hours of live-cell imaging and typically lasted five to six hours. Are the single-time point interphase snapshots shown here able to capture events at this frequency?

We believe the reviewer is referring to Yi et al., Cancer Discovery, 2021 (Figure 3F). This is an important study of live-cell labelling of ecDNA with which comparison is necessarily undertaken throughout the manuscript. We would still expect to observe clustering (<200nm) in the data presented within our manuscript if transcriptional-range clustering was occurring in interphase nuclei. We have added a sentence to clarify the method selected here (Discussion, para 3).: ‘…using Ripley’s K to objectively call instances of significant clustering at given distances from ecDNA x,y,z coordinates.’

In addition, we note fixed-cell analysis of ecDNA hub formation in Hung et al., Nature, 2021 in a range of cell lines (Figure 1), which we have included as a central reference point throughout the text, and highlighting cell lines of particular reference to this study (See Discussion). This gives us further confidence that fixed cell imaging is an appropriate modality to study whether ecDNA clustering is a key phenomenon in ecDNA gene regulation.

Further, it may be difficult to generalise the timeline of ecDNA hub formation. The PC3 cell line (live-cell imaging in Yi et al., 2021), along with other established cell lines studied in the context of ecDNA, may have inherent differences, such as cell cycle length, mitotic index, physiology, growth characteristics etc in comparison to primary GBM cells which might affect these frequencies. We chose to use patient-derived primary GBM cultures that mirror human disease, and have focused on quantitative spatial analysis. We have explored this in the Discussion, para 4.

5) ecDNA hubs show preferential colocalization with RNAPII and therefore, as these results do not find ecDNA hubs, it is not surprising that there was no colocalization between ecDNA molecules and RNAPII.

We have amended the first sentence of Results section three to make this clearer: ‘We next assessed whether ecDNA foci, which, whilst not clustered, share regional localization…’. It remained important to evaluate if this regional localisation was associated with RNAPol2, as this might have suggested that, while not in ecDNA hubs, ecDNA were instead associated with large RNAPol2 foci of the nucleus in ecDNA/RNAPII hubs.

6) The E26 EGFR line (Figure 2D) as well as the E25 CDK4/PDGFRA ecDNA line shows evidence of clustering of molecules (Figure 3D and 3E). The measuring method of averaging distances would undervalue those as the signal is drained in a nucleus with many ecDNA molecules.

We have supplied a sample of single slice images of E25, E26 and E28 nuclei, and of the E20 nuclei shown without (A) and with (B) colocalisation for review. We selected maximum intensity projection (MIP) images for the main figures for ease of visualisation of these small foci and the figure legend notes that the images provided are maximum intensity projections of the analysed 3D Z-stacks. As a result, this may be seen as clustering if two foci align closely in the z plane.

In addition to this, two or more foci may be closely located by chance, particularly when there are many foci in a single nucleus. Accounting for this chance through mathematical modelling is the basis of the Ripley’s K tool, as this allows number of foci and size of volume to be accounted for. While we did not observe clustering at distances in keeping with transcriptional contacts, we did observe regular non-random distribution of loci within nuclei at >300nm distances. We propose an alternative hypothesis for ecDNA distribution that does not relate to coordinated transcription.

We acknowledge the concerns expressed about averaging distances. To give a more complete picture, we have therefore included Cumulative Frequency Distribution (CFD) graphs, which include the shortest distance between all the indicated foci, measured across all nuclei measured, to show clearly the distribution of all available data. These are in Figure 2 and Figure 3 for all cell lines analysed.

We agree that the averaging data can only give an overall impression. Indeed, for a nucleus with many foci, the mean shortest interprobe distances would in fact be smaller. This is intended to give an overall picture of ecDNA spatial relationships, which is then quantified in detail using the Ripley’s K function analysis. This is clearly explained for the reader. We have therefore retained the mean shortest and single shortest foci data in the Figure 1 – supplemental figure 1 – and amended the results text to give greater clarity.

A general comment – debate on data and interpretation is a critical part of science. The paper has value in stimulating debate. However, the oppositional tone is distracting. The paper would be more effective and impactful, at least in this reviewer's view, if it were written from the standpoint of trying to understand the complex dynamics of ecDNA biology.

We agree an oppositional tone is unhelpful. Our intention was very much to present our findings in an objective manner with reference to key previous studies. We have made adjustments throughout the text to stimulate debate as to why discrepant conclusions have been reached. We did not feel these were unfairly critical or partisan – but merely our attempt to explain why different results have been obtained in our study.

Reviewer #3 (Significance (Required)):Please see above.

Author note:

Further amendments:

Recent work proposing that ecDNA act as mobile super-enhancers for chromosomal targets has raised the possibility that ecDNA can actively recruit RNA PolII to drive ‘ecDNA-associated phase separation’ (Zhu et al. 2021). – This previously included a reference to Adleman and Martin, 2021, but on further review have edited this to the primary reference only as shown here.

We noted an error in para 4 of the discussion:

‘…results in large (approx. 1.75Mb) ecDNA (Hung et al., 2021; Wu et al., 2019)’.

This should have read ‘1.75 μm’, and so we have updated this as follows:

‘…results in large (4.328Mb, approx. 1.75μm diameter) ecDNA (Hung et al., 2021; Wu et al., 2019).